# RHYTHM: Reasoning with Hierarchical Temporal Tokenization for Human Mobility

**Haoyu He**[†]    **Haozheng Luo**[‡]    **Yan Chen**[‡]    **Qi R. Wang**[†]

[†] Northeastern University    [‡] Northwestern University
{he.haoyu1, q.wang}@northeastern.edu
hluo@u.northwestern.edu, ychen@northwestern.edu

## Abstract

Predicting human mobility is inherently challenging due to complex long-range dependencies and multi-scale periodic behaviors. To address this, we introduce **RHYTHM** (Reasoning with Hierarchical Temporal Tokenization for Human Mobility), a unified framework that leverages large language models (LLMs) as general-purpose spatio-temporal predictors and trajectory reasoners. Methodologically, RHYTHM employs temporal tokenization to partition each trajectory into daily segments and encode them as discrete tokens with hierarchical attention that captures both daily and weekly dependencies, thereby quadratically reducing the sequence length while preserving cyclical information. Additionally, we enrich token representations by adding pre-computed prompt embeddings for trajectory segments and prediction targets via a frozen LLM, and feeding these combined embeddings back into the LLM backbone to capture complex interdependencies. Computationally, RHYTHM keeps the pretrained LLM backbone frozen, yielding faster training and lower memory usage. We evaluate our model against state-of-the-art methods using three real-world datasets. Notably, RHYTHM achieves a **2.4%** improvement in overall accuracy, a **5.0%** increase on weekends, and a **24.6%** reduction in training time. Code is publicly available at https://github.com/he-h/rhythm.

## 1 Introduction

Human mobility shapes transportation systems [6, 62], informs epidemic control strategies [63, 8], and guides sustainable city planning [4], making accurate movement prediction essential for optimizing infrastructure, managing disease spread, and building resilient communities [42]. Yet human trajectories exhibit long-range dependencies, spatial heterogeneity [78, 19], and dynamic influences such as weather anomalies or special events [7, 36], producing non-stationary, multi-scale spatio-temporal patterns.

To address this challenge, we introduce **RHYTHM** (Reasoning with Hierarchical Temporal Tokenization for Human Mobility), a human mobility foundation model that rethinks mobility modeling via structured temporal abstraction. We posit that human mobility, like language, follows compositional structures: daily routines form tokens, and weekly rhythms form higher-order syntax. RHYTHM operationalizes this analogy by tokenizing time into segments and reasoning over them with a frozen large language model (LLM). This framework unites multi-scale temporal tokenization with the reasoning capabilities of pretrained LLMs, delivering a scalable and general approach for mobility prediction with reduced computational cost.

The design of RHYTHM is inspired by inherent patterns of human mobility. *People's movements are not random; they follow an underlying order marked by recurring daily and weekly rhythms* [12, 24, 18]. Notably, Song et al. [58] quantify this regularity by showing that 93% of daily trajectories are

predictable, underscoring the critical role of cyclical temporal context in mobility modeling. Capturing this cyclical regularity requires models that can jointly represent local behaviors (e.g., morning commutes) and global temporal dependencies (e.g., weekly routines). Yet existing approaches fall short: Markov and RNN-based methods either disregard long-term periodicity or suffer vanishing gradients over long sequences [76, 19, 22], while transformer-based methods treat time as static, failing to disentangle multi-scale temporal patterns [27, 72]. To bridge this gap, we decompose each trajectory into meaningful segments, tokenizing each into discrete representations that capture local patterns through intra-segment attention. These segment tokens are then pooled into higher-level representations, enabling inter-segment attention to model long-range dependencies across days, as illustrated in Figure 1, thereby reducing sequence length and quadratically lowering attention cost. Each segment token is augmented with pre-computed semantic embeddings derived from a frozen LLM, enriching temporal tokens with contextual meaning before being processed by the LLM backbone for reasoning.

Recent advances demonstrates LLMs' remarkable capabilities not only as sequential representation extractors to capture the spatio-temporal attention patterns but also as reasoning models [13, 9]. Prior works [25, 59, 17, 21] demonstrate their reasoning capabilities through techniques such as few-shot prompting [9], chain-of-thought reasoning [50, 49, 66], and in-context learning [16]. However, mobility-specific models like PMT [71] and ST-MoE-BERT [26] lack the capability to leverage LLMs for modeling complex correlations in human flows, limiting

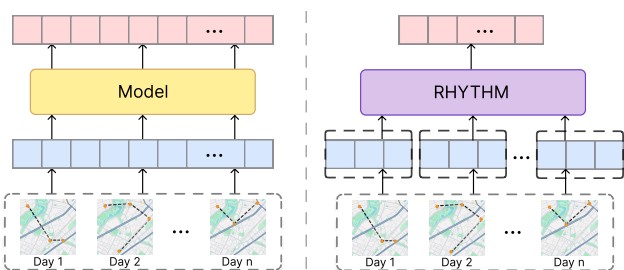

Figure 1: **Motivation for RHYTHM.** Instead of processing entire trajectories as a continuous sequence, RHYTHM segments them into tokens to better capture periodic patterns.

their predictive performance. By integrating an LLM-based reasoning module, RHYTHM more effectively models these complex interdependencies. To maintain scalability, RHYTHM adopts a parameter-efficient adaptation strategy by freezing the pretrained LLM and avoid extensive fine-tuning. This design captures fine-grained spatio-temporal dynamics, deep semantic context, and leverages LLM reasoning—all while minimizing computational and memory overhead—making RHYTHM ideally suited for deployment in resource-constrained, real-world environments.

**Contributions.** We propose **RHYTHM**, a unified, computationally efficient framework that captures both temporal dynamics and cyclical patterns, as illustrated in Figure 2. Our contributions are as follows:

- We introduce temporal tokenization that encodes daily mobility patterns as discrete tokens, reducing the processed sequence length while capturing cyclical and multi-scale mobility dependencies through hierarchical attention mechanism.

- We design an efficient prompt-guided approach that integrates semantic trajectory information and task description with segment embeddings, enhancing RHYTHM's ability to interpret complex mobility patterns.

- We propose a parameter-efficient adaptation strategy using frozen pretrained LLMs, reducing trainable parameters to **12.37%** of the full model size and achieving a **24.6%** reduction in computational cost compared to other baselines.

- Empirically, we evaluate RHYTHM on three real-world mobility datasets, demonstrating superior performance compared to state-of-the-art models. RHYTHM achieves a **2.4%** improvement in prediction accuracy, with a **5.0%** increase on weekends.

## 2 Related Work

In this section, we provide a brief overview of related work, with a detailed review in Appendix B.

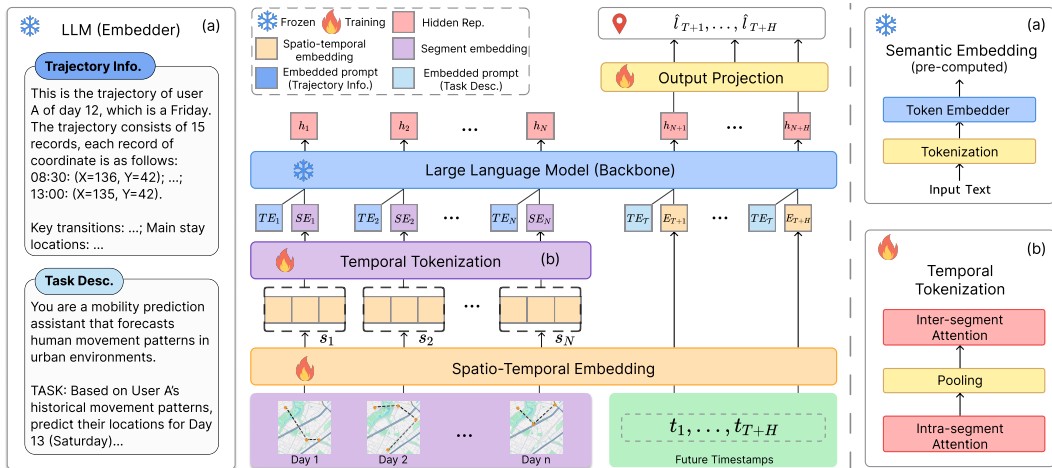

Figure 2: **The proposed architecture of RHYTHM.** Our framework processes historical trajectories through spatio-temporal embedding and temporal tokenization (**b**), capturing local and global dependencies via hierarchical attention. Segment representations are enriched with semantic embeddings from trajectory information, while future timestamps incorporate task description context (**a**). This combined sequence passes through a frozen LLM backbone with output projection to generate location predictions.

**Mobility Prediction.** Human mobility prediction progresses from probabilistic approaches [75, 22] to deep learning architectures, as demonstrated in recent studies. Sequence models like LSTM [37] and attention mechanisms [19] improve temporal modeling, while graph-based methods [55, 14] integrate spatial relationships. Transformer architectures [65, 77, 45] further enhance long-range dependency modeling but struggle with the hierarchical structure of mobility patterns. Recent work with LLMs [20, 64] shows promise but typically treats mobility as generic sequences, ignoring the inherent periodicity of human movement.

**Cross-domain Adaptation of LLMs.** LLMs emerge as powerful natural language processing systems and quickly evolve into general-purpose foundation models capable of reasoning and generation tasks [9, 1]. Their remarkable adaptability has enabled successful applications in computer vision [5, 53], speech [69, 46], biomedicine [82, 44, 57], time series forecasting [11, 81], and finance [31, 70]. While many adaptations rely on parameter-efficient fine-tuning methods like LoRA [29], recent approaches maintain frozen LLMs by utilizing them as sequential representation extractors, preserving their semantic capabilities while reducing computational costs [40, 32, 2]. To the best of our knowledge, RHYTHM is the **first** approach that adapts frozen LLMs to mobility prediction without compromising the model's reasoning capabilities or requiring extensive fine-tuning.

## 3 Method

In this section, we introduce **RHYTHM**, an LLM-based deep architecture tailored for prompt-guided representation learning of spatio-temporal patterns with its periodicity, as shown in Figure 2. In the following, we first define the problem and then introduce the model structure of RHYTHM, including its computational efficiency and theoretical guarantees.

### 3.1 Problem definition

Let $\mathcal{X} = \{x_1, x_2, \ldots, x_T\}$ denote a user's historical trajectory, where each $x_i = (t_i, l_i)$ consists of a timestamp $t_i$ and a location $l_i \in \mathcal{L}$ from a finite set of locations $\mathcal{L}$. Given a sequence of future timestamps $\mathcal{T} = \{t_{T+1}, t_{T+2}, \ldots, t_{T+H}\}$ with prediction horizon $H$, the goal is to predict the corresponding future locations $\mathcal{Y} = \{l_{T+1}, l_{T+2}, \ldots, l_{T+H}\}$. Formally, we seek a function

$$f : (\mathcal{X}, \mathcal{T}) \mapsto \mathcal{Y},$$

which maps historical trajectories and future timestamps to the user's future locations.

## 3.2 Model structure

**Spatio-Temporal Feature Encoding.** For each observation $x_i$, we construct temporal embeddings to capture cyclical patterns in human movements:

$$\mathbf{E}_i^{\text{temporal}} = \mathbf{E}^{\text{ToD}}(t_i) \| \mathbf{E}^{\text{DoW}}(t_i),$$

where $\cdot \| \cdot$ indicates concatenation, $\mathbf{E}^{\text{ToD}}$ represents the time-of-day embedding (capturing 24-hour cycles), and $\mathbf{E}^{\text{DoW}}$ represents the day-of-week embedding (capturing weekly patterns). These are learnable embeddings that map discrete temporal indices to continuous representations, with $\mathbf{E}_i^{\text{temporal}} \in \mathbb{R}^D$ with $D$ matching the backbone LLM's input dimension.

Spatial embeddings $\mathbf{E}_i^{\text{spatial}} \in \mathbb{R}^D$ for location $l_i$ is defined as:

$$\mathbf{E}_i^{\text{spatial}} = \mathbf{E}^{\text{Loc}}(l_i) \| (W_{\text{coord}}[\text{lat}_i, \text{lon}_i]^T + b_{\text{coord}}),$$

where $\mathbf{E}^{\text{Loc}}$ denotes the categorical location embedding, and the second term projects the geographic coordinates $(\text{lat}_i, \text{lon}_i)$ into the embedding space. Here, $W_{\text{coord}} \in \mathbb{R}^{d_{\text{coord}} \times 2}$ is the projection matrix and $d_{\text{coord}}$ denotes the projected dimension.

The spatio-temporal embedding $\mathbf{E}_i \in \mathbb{R}^D$ is obtained by element-wise addition:

$$\mathbf{E}_i = \mathbf{E}_i^{\text{temporal}} + \mathbf{E}_i^{\text{spatial}}.$$

For future timestamps without known locations or missing historical records, the spatial component is set to zero, allowing the model to operate on temporal information alone while preserving dimensional consistency.

**Temporal Tokenization.** Human mobility patterns exhibit inherent multi-scale temporal structures that span both short-term routines (e.g., daily activities) and long-term periodicities (e.g., weekly rhythms) [58, 24]. To effectively model these dynamics, RHYTHM employs a temporal tokenization mechanism that effectively disentangles local patterns from global dependencies, inspired by Liu et al. [40]. Formally, we partition the embedded sequence $\mathcal{X}$ into $N$ non-overlapping segments $\{\mathbf{s}_1, \mathbf{s}_2, \ldots, \mathbf{s}_N\}$, each capturing meaningful temporal intervals (e.g., daily patterns):

$$\mathbf{s}_i = \{E_{(i-1)L+1}, E_{(i-1)L+2}, \ldots, E_{iL}\} \quad \text{for } i = 1, 2, \ldots, N,$$

where each segment $\mathbf{s_i}$ has length $L$ (number of time steps). Within each segment, we employ intra-segment attention to model local temporal dependencies:

$$\widetilde{\mathbf{E}}^{(i)} = \text{Attention}(\mathbf{s}_i).$$

Our attention mechanism follows a pre-norm transformer architecture with a gated feed-forward network as introduced by Dubey et al. [17], which enhances gradient flow during training and increases model expressivity. The implementation details are provided in Appendix C. To enable efficient modeling of cross-segment dependencies, we apply a learnable pooling operation that condenses each segment into a discrete token representation:

$$\mathbf{SE}_i = \text{Pool}\big(\widetilde{\mathbf{E}}^{(i)}\big).$$

The resulting segment tokens $\{\widetilde{\mathbf{SE}}_1, \widetilde{\mathbf{SE}}_2, \ldots, \widetilde{\mathbf{SE}}_N\}$ undergo inter-segment attention to capture broader temporal context and long-range dependencies:

$$\widetilde{\mathbf{SE}}_{1:N} = \text{Attention}(\mathbf{SE}_{1:N}),$$

yielding refined segment-level embedding $\widetilde{\mathbf{SE}_i} \in \mathbb{R}^D$ that integrates contextual information across multiple temporal scales. By reducing the effective sequence length from $T$ to $N$ while preserving both fine-grained temporal dynamics and long-term dependencies, our approach addresses the computational challenges of modeling extended mobility trajectories.

**Semantic Context Integration.** Prior work such as FPT [81] employs LLMs as general-purpose sequential representation extractors. However, models like those in Wu et al. [67], Nie et al. [48] typically discard semantic embeddings and other critical information. For instance, timestamp attributes (e.g., day of the week, hour of the day) are essential for capturing chronological patterns in

human mobility, while spatial details—such as coordinates—provide additional context for accurate prediction.

While recent works [40, 68, 80] begin to incorporate semantic information in sequential data, they typically employ traditional embedding approaches that fail to holistically capture the rich contextual information inherent in mobility patterns. Our work addresses this limitation by developing a mobility-based semantic embedding method that leverages detailed trajectory information. A key challenge in utilizing LLMs for mobility prediction is balancing information richness with computational efficiency. Unlike approaches such as LLM-Mob [64] that rely on extensive prompting—which can lead to excessive context lengths and computational overhead, our approach breaks trajectory information into smaller, segment-specific pieces that retain essential mobility patterns while ensuring shorter prompts for each component. This decomposition significantly improves computational efficiency without sacrificing semantic richness. For each segment token, we provide informative trajectory descriptions, and for each future timestamp, we provide task descriptions and timestamp information, clarifying the expected output as shown in Appendix D. These prompts are then processed by pretrained LLMs to generate semantic embeddings. We adopt a strategy that uses the special end-of-sequence (<EOS>) token for positional embedding, thereby integrating semantic information without extending the overall context length.

Technically, we define the semantic embedding $\mathbf{TE}_i \in \mathbb{R}^D$ for each token as follows:

$$\mathbf{TE}_i = \text{SelectLast}(\text{LLM}(\text{Prompt}(x_{(i-1)L+1:iL}))).$$

Similarly semantic embedding of task description for future timestamp is defined as $\mathbf{TE}_{\mathcal{T}} = \text{SelectLast}(\text{LLM}(\text{Prompt}(\mathcal{T})))$. Notably, $\mathbf{TE}$ is pre-computed using the LLM, so a runtime forward pass through the language model is not required during training.

**Semantic–Temporal Alignment for Mobility Prediction.** Since the latent space of the LLM encompasses both temporal tokens and semantic tokens, the semantic embedding can be aligned with the corresponding time span without extending the context length. Consequently, the combined embedding $\mathbf{CE}_i$ for segment $i$ is obtained by elementwise adding the segment embedding $\widetilde{\mathbf{SE}}_i$ and semantic embedding $\mathbf{SE}_i$:

$$\mathbf{CE}_i = \widetilde{\mathbf{SE}}_i + \mathbf{TE}_i.$$

Here, $\mathbf{TE}_i$ serves a role similar to positional embeddings [61], while avoiding the sequence length overhead incurred by prompt concatenation [32]. Similarly, the combined embedding for future timestep $T + j$ is computed as $\mathbf{CE}_{N+j} = \widehat{\mathbf{E}}_{N+j} + \mathbf{TE}_{\mathcal{T}}$, following the combined embedding for $N$ segments.

After obtaining the enriched embeddings $\mathbf{CE}_i$, we feed them into the backbone of RHYTHM–a pretrained LLM. The LLM processes these embeddings through its deep layers, performing in-context reasoning over the aligned temporal and semantic information, and yields contextualized hidden representations $h_i$ from its last hidden layer.

$$h_i = \text{LLM}(\mathbf{CE}_i).$$

Then we apply an output projection layer to map the LLM's final representations to a set of logits corresponding to candidate locations.

$$P(l_{T+j}|\mathcal{X}, \mathcal{T}) = \text{softmax}(W_o \mathbf{h}_{N+j} + \mathbf{b}_o),$$

where $W_o \in \mathbb{R}^{|\mathcal{L}| \times D}$. These logits are then used to determine the most likely location predictions, thereby generating human mobility forecasting as defined in our problem statement. See Appendix E for implementation details.

### 3.3 Computational Efficiency

RHYTHM achieves computational and parameter efficiency through several complementary design choices. Semantic embeddings are computed once—offline—using the frozen LLM prior to model training, thereby eliminating any need for language-model inference at runtime. Simultaneously, our temporal tokenization mechanism significantly reduces sequence length from $T + H$ to $N + H$, thereby decreasing the quadratic attention complexity from $\mathcal{O}((T + H)^2)$ to $\mathcal{O}((N + H)^2)$, which is particularly valuable when processing extended mobility histories. Furthermore, we keep the LLM

backbone parameters frozen during training, resulting in faster convergence and reduced memory requirements. The combination of these approaches enables RHYTHM to efficiently process long mobility trajectories while maintaining strong predictive performance (as shown in Figure 3), making it suitable for large-scale mobility prediction tasks where computational resources may be constrained.

## 3.4 Theoretical Guarantee

We emphasize that our design choices provide strong theoretical guarantees. Employing an LLM as a universal sequential representation extractor provides two key advantages: (1) it ensures the convergence of output values, as demonstrated in Zhou et al. [81, Theorem E.2], and (2) it guarantees a uniform distribution of the feature space in the last hidden layer of the LLM, as outlined in Zhou et al. [81, Theorem E.3]. Together, these properties enable LLMs to enhance the learning capability of the final multi-layer perceptron layer. Additionally, since our model is transformer-based, Ramsauer et al. [54] demonstrates that the transformer architecture is a special case of modern Hopfield networks. Our approach has a guaranteed upper bound on memory retrieval error in LLMs [30, Lemma 3.2]. These theoretical benefits reinforce our method, and our results provide validation.

## 4 Experiment

In this section, we conduct experiments to demonstrate the performance and efficiency of RHYTHM. We evaluate the performance of RHYTHM on three real-world mobility datasets and compare it with several state-of-the-art baselines. We also conduct a series of ablation studies to investigate the effectiveness of the proposed strategies.

**Models.**   To evaluate the model's performance on mobility prediction, we use multiple pretrained LLMs as the backbone of RHYTHM. These models are obtained from Hugging Face along with their pretrained weights. The specific LLM variants used are detailed in Appendix G.4.

**Evaluation Metrics.**   For mobility prediction, we employ Accuracy@k, where candidate locations are ranked based on model-predicted probabilities, and a prediction is considered correct if the true location is among the top k—and Mean Reciprocal Rank (MRR) to evaluate ranking performance. These metrics have been shown to correlate well with human mobility prediction tasks [23, 19]. We also utilize Dynamic Time Warping (DTW) [47] and BLEU [51] as real-world metrics to evaluate the performance. DTW quantifies their spatial alignment, while BLEU measures the n-gram similarity between predicted and ground-truth trajectories. Detailed descriptions about the evaluation metrics can be found in Appendix G.1.

**Datasets.**   We evaluate our approach on three real-world datasets collected from the cities of Kumamoto, Sapporo, and Hiroshima sourced from YJMob100K [74]. Each day is divided into 48 time slots (each representing 30 minutes), though not every slot contains an observation. Each dataset is divided into training, validation, and test sets based on days, with 70%, 20%, and 10% of the data allocated to each set, respectively. More details about the dataset can be found in Appendix F.

**Settings.**   The temporal resolution is 30 minutes. In our experiment, we use a 7-day lookback window with 336 time slots and set the prediction horizon to 48 time slots (1 day). Also, we set the segment length as 48 time slots for our experiments. The model is trained using cross-entropy loss to maximize prediction accuracy across the target locations.

## 4.1 Overall Performance

To assess the efficiency of RHYTHM on human mobility prediction, we compare RHYTHM with several state-of-the-art baselines. In this experiment, we evaluate the models on the test datasets using FP16 precision. We conduct each evaluation three times with different random seeds and present the average for each metric.

**Baselines.**   To evaluate the performance of RHYTHM, we compare to LSTM-based models, transformer-based models, and LLM-based models. For the transformer-based models, we conduct experiments with PatchTST [48], PMT [71], ST-MoE-BERT [26], CMHSA [27], iTransformer [39]

and COLA [65]. Among these models, ST-MoE-BERT, PMT and COLA are the state-of-the-art models for human mobility prediction. PatchTST and iTransformer are two powerful transformer-based models for time series forecasting. We add the spatiotemporal embedding to the input of those transformer-based time-series models for fair comparison. For the LLM-based models, we conduct experiments with Time-LLM [32] and Mobility-LLM [23]. Time-LLM is a state-of-the-art model for time series forecasting using LLMs. We also add the spatiotemporal embedding to the input of Time-LLM for fair comparison. Additionally, in order to make a fair comparison, we use the Llama-3.2 1B[1] as the pretrained LLM model for fine-tuning Time-LLM. Mobility-LLM is a versatile LLM-based framework designed for multiple mobility tasks. For the LSTM-based models, we conduct experiments with LSTM [34] and DeepMove [19].

**Results.** Table 1 show that RHYTHM outperforms the baselines across three datasets in most metrics. On the Sapporo and Hiroshima dataset, RHYTHM achieves the best performance in all evaluation metric. These findings underscore the effectiveness of RHYTHM in mobility prediction tasks. CMHSA and PMT may perform better in Accuracy@3 on Kumamoto due to their specialized attention mechanisms that effectively capture mid-range candidate locations in this region. Despite sharing an LLM-based architecture, Mobility-LLM underperforms compared to RHYTHM, likely because it was primarily designed for visiting intention tasks requiring rich semantic context. In contrast, RHYTHM leverages temporal tokenization and LLM to model multi-scale spatio-temporal dependencies, prioritizing precise location likelihood maximization. This design focus enables RHYTHM to excel in top-rank precision metrics. Overall, RHYTHM achieves a 2.4% improvement in Accuracy@1 and a 1.0% Accuracy@5 respectively compared to the best baseline model.

Table 1: **Performance of RHYTHM and baselines on the Kumamoto, Sapporo, and Hiroshima datasets.** The evaluation metrics include Accuracy@k for different values of k. The reported results are averaged over three runs; variance values are omitted as all are $\leq 2\%$. The best results are highlighted in **bold**, and the second-best results are underlined. RHYTHM demonstrates superior performance compared to baselines across most configurations.

| Model | Kumamoto | | | Sapporo | | | Hiroshima | | |
|---|---|---|---|---|---|---|---|---|---|
| | Acc@1 | Acc@3 | Acc@5 | Acc@1 | Acc@3 | Acc@5 | Acc@1 | Acc@3 | Acc@5 |
| LSTM | 0.2652 | 0.4799 | 0.5472 | 0.2310 | 0.3940 | 0.4526 | 0.2129 | 0.3775 | 0.4415 |
| DeepMove | 0.2779 | 0.4986 | 0.5683 | 0.2825 | 0.4672 | 0.5264 | 0.2804 | 0.4810 | 0.5477 |
| PatchTST | 0.2751 | 0.5018 | 0.5716 | 0.2703 | 0.4582 | 0.5168 | 0.2752 | 0.4839 | 0.5522 |
| iTransformer | 0.2609 | 0.4724 | 0.5412 | 0.2696 | 0.4500 | 0.5070 | 0.2804 | 0.4857 | 0.5523 |
| Time-LLM | 0.2712 | 0.4848 | 0.5535 | 0.2792 | 0.4746 | 0.5352 | 0.2698 | 0.4753 | 0.5426 |
| CMHSA | 0.2862 | 0.5182 | 0.5887 | 0.2890 | **0.4901** | 0.5525 | 0.2874 | 0.5001 | 0.5684 |
| PMT | 0.2697 | 0.4475 | 0.5187 | 0.2878 | 0.4896 | 0.5522 | 0.2850 | 0.4982 | 0.5668 |
| COLA | 0.2864 | 0.5186 | 0.5896 | 0.2847 | 0.4865 | 0.5497 | 0.2874 | 0.5013 | 0.5708 |
| ST-MoE-BERT | 0.2862 | 0.5155 | 0.5871 | 0.2869 | 0.4856 | 0.5480 | 0.2839 | 0.4925 | 0.5601 |
| Mobility-LLM | 0.2666 | 0.4793 | 0.5448 | 0.2838 | 0.4703 | 0.5288 | 0.2826 | 0.4856 | 0.5525 |
| RHYTHM-Llama-1B | 0.2929 | 0.5200 | 0.5835 | 0.2931 | 0.4876 | 0.5502 | 0.2913 | 0.5027 | 0.5753 |
| RHYTHM-Gemma-2B | 0.2923 | 0.5191 | 0.5932 | **0.2943** | 0.4896 | **0.5545** | **0.2953** | **0.5074** | **0.5798** |
| RHYTHM-Llama-3B | **0.2941** | **0.5205** | **0.5947** | 0.2938 | 0.4875 | 0.5523 | 0.2929 | 0.5032 | 0.5756 |

**Geographical Evaluation.** We evaluate RHYTHM against baseline models using BLEU and DTW, which respectively measure n-gram similarity and spatial alignment error between predicted and ground-truth trajectories. As shown in Table 2, RHYTHM scores the best DTW performance on Sapporo, demonstrating superior spatial alignment. While COLA leads in BLEU scores for all cities, RHYTHM ranks second in Kumamoto. This highlights a key trade-off between exact sequence matching and minimizing spatial deviations. One potential explanation is that COLA employs a post-hoc adjustment technique that recalibrates predictions to better align with the long-tail frequency distribution of locations, which may enhance mid-tier accuracy by mitigating overconfidence in dominant locations. Notably, RHYTHM significantly outperforms LSTM-based methods and transformer baselines by leveraging temporal tokenization and prompt-guided reasoning to enhance sequential coherence and spatial precision. This results in an optimal balance for real-world mobility tasks. For MRR, RHYTHM consistently outperforms all baselines, achieving a 1.44% improvement over

---

[1]https://huggingface.co/meta-llama/Llama-3.2-1B

the best baseline and demonstrates its superior ranking capability across diverse mobility patterns. Additional experimental results and extended evaluations are reported in Appendix H.

Table 2: **Performance comparison of RHYTHM with baselines using geographical metrics.** The evaluation metrics include DTW (↓), BLEU (↑), and MRR (↑). The best results are highlighted in **bold**, and the second-best results are underlined.

| Model | Kumamoto | | | Sapporo | | | Hiroshima | | |
|---|---|---|---|---|---|---|---|---|---|
| | DTW | BLEU | MRR | DTW | BLEU | MRR | DTW | BLEU | MRR |
| LSTM | 5014 | 0.1564 | 0.3860 | 4507 | 0.1716 | 0.3270 | 5908 | 0.1544 | 0.3113 |
| DeepMove | 4630 | 0.1746 | 0.4021 | 3818 | 0.1959 | 0.3887 | 4981 | 0.1933 | 0.3959 |
| PatchTST | 5251 | 0.1315 | 0.4021 | 4099 | 0.1784 | 0.3773 | 5021 | 0.1884 | 0.3945 |
| iTransformer | 6178 | 0.1275 | 0.3796 | 4074 | 0.1780 | 0.3730 | 5094 | 0.1789 | 0.3977 |
| Time-LLM | 5984 | 0.1285 | 0.3912 | 3915 | 0.2145 | 0.3902 | 5126 | 0.1988 | 0.3872 |
| CMHSA | 4490 | 0.1810 | 0.4158 | 3786 | 0.2299 | 0.4034 | 4841 | 0.2289 | 0.4086 |
| PMT | 4536 | 0.1524 | 0.3720 | 3799 | 0.2017 | 0.4026 | 4851 | 0.2009 | 0.4065 |
| COLA | 4446 | **0.2064** | 0.4164 | 3793 | **0.2496** | 0.3996 | **4840** | **0.2445** | 0.4095 |
| ST-MoE-BERT | 4691 | 0.1557 | 0.4151 | 3796 | 0.2102 | 0.4001 | 4889 | 0.2117 | 0.4031 |
| Mobility-LLM | 5603 | 0.1649 | 0.3858 | 3911 | 0.1917 | 0.3902 | 4985 | 0.2056 | 0.3990 |
| RHYTHM-Llama-1B | 4478 | 0.1793 | 0.4216 | **3745** | **0.2496** | 0.4045 | 5059 | 0.2083 | 0.4069 |
| RHYTHM-Gemma-2B | **4416** | 0.1928 | 0.4205 | 3995 | 0.2019 | **0.4065** | 4857 | 0.2109 | **0.4173** |
| RHYTHM-Llama-3B | 4470 | 0.1814 | **0.4220** | 4035 | 0.1917 | 0.4048 | 4935 | 0.2093 | 0.4140 |

**Transferability.**    To demonstrate that RHYTHM transfers well across pretrained LLMs, we vary the size of the pretrained backbone and train it on the mobility prediction datasets (see Table 1 for detailed results). In our experiments, we change the size of the pretrained model in RHYTHM and test them on the mobility prediction datasets. We use the Llama-3.2-1B, Llama-3.2-3B, and Gemma-2-2B model as the pretrained backbone of RHYTHM. The results indicate that the performance of RHYTHM improves as the model size increases. Notably, Llama-3.2-3B and Gemma-2-2B model outperforms the Llama-3.2-1B model in most metrics. This result demonstrates the performance of RHYTHM scales with LLM size and suggests that larger models may achieve even greater performance improvements on larger datasets. Note that our models are pretrained with 30 epochs. It's plausible that Llama-3.2-3B model requires more epochs to fully converge and realize its full performance potential compared with Llama-3.2-1B model. However, Llama-3.2-3B model still achieves competitive performance compared to Llama-3.2-1B model. Overall, Llama-3.2-3B model demonstrated a 0.40% improvement in Acc@1 compared to Llama-3.2-1B model, highlighting the scalability of RHYTHM.

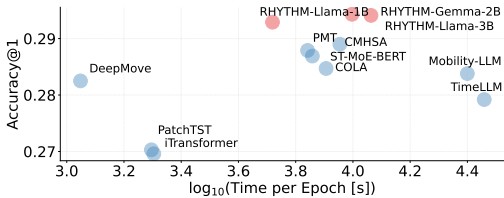

Figure 3: **Training Speed vs. performance of RHYTHM and baseline models on the Sapporo Dataset.**

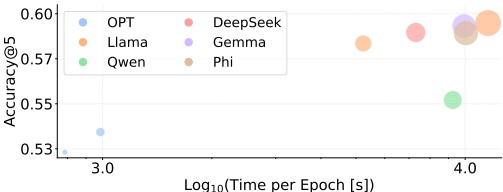

Figure 4: **Efficiency comparison of alternative LLMs, evaluated by the same configuration of Table 4.**

**Training Speed.**    To evaluate the training speed of RHYTHM, we conduct experiments on Sapporo using the same training configuration. We run these experiments on a single NVIDIA A100 GPU with 40GB of memory. The results are shown in Figure 3. RHYTHM reduces the number of trainable parameters to only 12.37% of the full model size, reflecting its parameter-efficient design. In terms of runtime, it achieves a 24.6% reduction in training time compared with the best-performing baseline, while remaining faster than most other models, being 80.6% faster than LLM-based methods on average. Although RHYTHM is slower than lightweight models such as LSTM, DeepMove, PatchTST, and iTransformer, it substantially outperforms them in accuracy. Moreover, RHYTHM maintains computational efficiency comparable to PMT, COLA and ST-MoE-BERT, despite having

significantly higher parameter counts, demonstrating its parameter-efficient design and scalable architecture. Furthermore, RHYTHM's training speed scales predictably with parameter count: Llama-3B is 2.2 times slower than Llama-1B model, while Gemma-2-2B shows a 1.9 times slowdown. A detailed breakdown of preprocessing time, storage cost, and training time across datasets is provided in Appendix I.

**Daily and Weekly Trend Analysis.** We analyze the periodic accuracy trends of RHYTHM and baselines on Sapporo, measuring performance fluctuations across daily and weekly intervals in Figure 5. RHYTHM demonstrates distinct performance characteristics: achieving 5.0% and 3.4% improvements during weekends and evening peak hours respectively, while showing comparable performance during highly regular periods like nighttime and standard weekday working hours. This pattern reveals a fundamental insight—RHYTHM excels precisely when mobility prediction becomes a complex decision-making task rather than simple pattern matching. During regular hours, mobility is largely deterministic with fixed routines where traditional models' pattern memorization suffices; however, weekends and transitional periods involve nuanced choices influenced by multiple contextual factors, aligning with findings from Barbosa et al. [3] on weekend variability. In these complex scenarios, RHYTHM's hierarchical attention captures both local daily context and global weekly patterns, while the LLM backbone provides reasoning capabilities to model non-routine decision points. This makes RHYTHM particularly valuable for real-world applications where handling irregular, unpredictable periods is crucial for system reliability, even if simpler models suffice for deterministic segments.

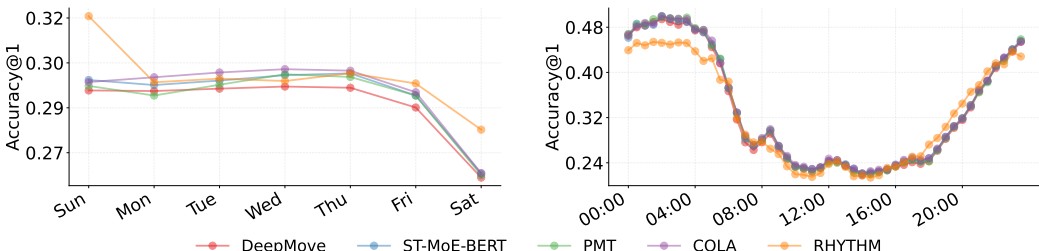

Figure 5: **Weekly (left) and daily (right) accuracy trends of RHYTHM and baselines on Sapporo.** These results illustrate that prediction performance fluctuates over both daily and weekly intervals.

## 4.2 Method Analysis

In this section, we perform ablation studies to assess the effectiveness of the proposed strategies and test the scaling behavior of RHYTHM.

**Ablation study.** All experiments utilize Llama-3.2-1B as the backbone model. To evaluate our key components, we conduct ablation studies across three datasets, as shown in Table 3. Removing temporal tokenization significantly degrades performance by 5.39%, while eliminating hierarchical attention (HA) results in a 0.90% decrease, demonstrating that structured temporal encoding is the most critical element of our framework. Regarding semantic enhancement, our findings indicate that both trajectory information and task description prompts contribute substantially to RHYTHM's effectiveness, with their removal causing a combined performance drop of 1.82%. Task descriptions yield marginally higher impact than trajectory information, with their removal causing an additional 0.10% performance decrease compared to omitting trajectory information. Further ablation studies examining the contribution of different design choices are included in Appendix J.

**Scaling Behavior.** Scalability is a critical factor in the success of large-scale models. We assess RHYTHM's scalability by analyzing its performance in different model sizes. We conduct experiments using pretrained LLMs of varying sizes, including OPT, Llama-3.2, DeepSeek-R1, Gemma-2, Phi-2, and Qwen 2.5 detailed in Table 6. As shown in Table 4, the predictor's prediction performance generally improves as the number of LLM parameters increases. This observation is consistent with scaling law dynamics in large models [33]. This scaling behavior highlights the trade-off between predictive performance and adaptation cost. To capture this balance, we assess RHYTHM's scalability across three dimensions: model performance, parameter size, and training and inference speed (time

Table 3: **Ablation study on each module in RHYTHM.** We evaluate each module's contribution to overall performance. The best results are highlighted in **bold**. Each module significantly influences RHYTHM's performance across all datasets.

| Model | Kumamoto | | | Sapporo | | | Hiroshima | | |
|---|---|---|---|---|---|---|---|---|---|
| | Acc@1 | Acc@3 | Acc@5 | Acc@1 | Acc@3 | Acc@5 | Acc@1 | Acc@3 | Acc@5 |
| RHYTHM | **0.2929** | **0.5200** | 0.5835 | **0.2938** | **0.4866** | **0.5502** | **0.2913** | **0.5027** | **0.5753** |
| w/o HA | 0.2917 | 0.5163 | 0.5881 | 0.2901 | 0.4856 | 0.5481 | 0.2895 | 0.4946 | 0.5657 |
| w/o token | 0.2801 | 0.5049 | 0.5764 | 0.2768 | 0.4775 | 0.5409 | 0.2749 | 0.4812 | 0.5535 |
| w/o Traj info. | 0.2914 | 0.5176 | **0.5891** | 0.2879 | 0.4842 | 0.5472 | 0.2858 | 0.4916 | 0.5633 |
| w/o Task desc. | 0.2895 | 0.5166 | 0.5889 | 0.2883 | 0.4839 | 0.5463 | 0.2882 | 0.4934 | 0.5648 |

per epoch), as shown in Figure 4. Our results indicate that the largest model, Llama-3.2-3B, achieves the best performance for human mobility prediction, while Llama-3.2-1B remains the most suitable choice in RHYTHM, providing an optimal balance between performance and computational cost.

Table 4: **Scalability Performance on RHYTHM.** We conduct experiments to evaluate the scalability of RHYTHM on Sapporo using pretrained models of varying parameter sizes. The evaluation metrics include Accuracy@k, MRR, training time per epoch (in seconds), and inference time per epoch (in seconds). The best results are highlighted in bold, while the second-best results are underlined. In most configurations, the performance of RHYTHM improves as the model size increases.

| Backbone | Training Time (s) | Inference Time (s) | Acc@1 | Acc@3 | Acc@5 | MRR |
|---|---|---|---|---|---|---|
| OPT-125M | 787 | 107 | 0.2798 | 0.4726 | 0.5231 | 0.3819 |
| OPT-350M | 986 | 224 | 0.2837 | 0.4789 | 0.5343 | 0.3923 |
| Llama-3.2-1B | 5235 | 359 | 0.2929 | 0.5200 | 0.5835 | 0.4216 |
| Qwen-2.5-1.5B | 9241 | 336 | 0.2897 | 0.4873 | 0.5521 | 0.4049 |
| DeepSeek-R1-1.5B | 7308 | 335 | 0.2921 | 0.5164 | 0.5896 | 0.4188 |
| Gemma-2-2B | 9928 | 559 | 0.2923 | 0.5191 | 0.5932 | 0.4205 |
| Phi-2 | 10047 | 693 | 0.2915 | 0.5166 | 0.5892 | 0.4183 |
| Llama-3.2-3B | 11566 | 762 | **0.2941** | **0.5205** | **0.5948** | **0.4220** |

## 5 Conclusion

This paper proposes RHYTHM, an efficient and scalable framework for mobility prediction. RHYTHM leverages temporal tokenization with hierarchical attention mechanisms to model spatio-temporal dependencies while incorporating semantic embeddings to capture cyclical patterns. The integration of frozen pretrained LLMs as reasoning engines enables RHYTHM to interpret nuanced decision-making processes that influence mobility choices, particularly in scenarios with irregular or non-routine movement patterns, at reduced computational costs. Empirical results demonstrate that RHYTHM significantly outperforms state-of-the-art methods in accuracy. Moreover, its high scalability allows for the seamless integration of different pretrained LLMs in a plug-and-play manner, offering a flexible and efficient prediction framework.

**Limitations.** It is worth noting that RHYTHM has certain limitations. Its performance depends heavily on the quality of pretrained LLMs, which were designed for language tasks rather than mobility prediction. If these models are resource-constrained, they may fail to capture user mobility patterns accurately. RHYTHM does not adopt an autoregressive prediction strategy; although widely studied in time-series modeling [40], we instead emphasize holistic sequence prediction to capture broader contextual dependencies. Future extensions may integrate autoregressive decoding to more closely mimic step-by-step human mobility decisions. In addition, while freezing pretrained LLMs improves efficiency, RHYTHM's training time remains high, limiting its practicality in some applications. Despite these challenges, RHYTHM provides a novel framework for mobility prediction, advancing efficiency and accuracy. Future work will focus on refining fine-tuning and quantization methods [43, 15, 73] to improve scalability and reduce resource demands.

## Acknowledgments and Disclosure of Funding

The authors would like to thank Mingzhen for insightful discussions and the anonymous reviewers for their constructive comments. H.H. and Q.R.W.'s work is supported by the National Science Foundation (NSF) under Grant Nos. 2125326, 2114197, 2228533, and 2402438, as well as by the Northeastern University iSUPER Impact Engine. H.L. is partially supported by the OpenAI Researcher Access Program. This research was supported in part through the computational resources and staff contributions provided by the Quest High Performance Computing facility at Northwestern University, which is jointly supported by the Office of the Provost, the Office for Research, and Northwestern University Information Technology. Any opinions, findings, conclusions, or recommendations expressed in the paper are those of the authors and do not necessarily reflect the views of the funding agencies.

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

# Supplementary Material

## A  Broader Impact

This paper introduces a new foundation model for human mobility, aiming to improve the reliability and generalization of foundation model applications in spatio-temporal domains. While the work does not have immediate societal implications, it lays the groundwork for future applications in urban planning, public health, disaster response, and transportation. However, the model may inadvertently encode or amplify biases present in the training data, potentially leading to inequitable outcomes in mobility predictions.

## B  Extended Related Work

**Mobility Prediction.**  Human mobility prediction has evolved from foundational statistical models to advanced deep learning frameworks. Physics-inspired models such as the gravity model [10] and the radiation model [56] predict aggregate population flows using distance and opportunity metrics but lack individual-level detail. To address this, probabilistic approaches like Markov chains [22] and tensor factorization [75] emerge, modeling location transitions at the user level. While these methods improve personalization, they struggle with sparse trajectories and higher-order dependencies inherent in real-world mobility data.

Deep learning introduces sequence-aware architectures like LSTM [37], which capture local temporal contexts, and attention-enhanced variants [19] that address vanishing gradients. However, these models often overlook cyclical patterns. Hybrid approaches like Graph-Flashback [55] and GC-DAN [14] integrate graph structures to model spatial relationships, but their reliance on fixed-length

sequences limits scalability for long-term forecasting. Transformers [61] revolutionize the field with self-attention mechanisms for long-term dependency modeling. Innovations like STAN [45] combine spatial-temporal attention for next-POI recommendation, while COLA [65] extends this to cross-city dynamics. GETNext [77] further refines predictions by disentangling individual preferences from population flows. Despite these advances, Transformer-based methods remain timestamp-centric, incurring quadratic complexity for multi-day sequences and failing to explicitly model hierarchical periodicities (e.g., daily vs. weekly rhythms).

While Transformer-based approaches improve long-term dependency modeling, they still rely on timestamp-centric encoding and struggle with hierarchical periodicities. Recent work explores large language models (LLMs) as an alternative, leveraging their strong generalization capabilities for mobility tasks [23, 38]. Studies like LLM-Mob [64] and AgentMove [20] leverage prompt engineering for next-location prediction and trajectory user linking, while TrajGPT [28] generates synthetic visits via autoregressive decoding. However, these approaches treat mobility sequences as generic token streams, neglecting structured periodic patterns and the modality gap between natural language and spatio-temporal data. Unlike existing LLM-based methods that treat mobility sequences as generic tokens, our approach uses temporal tokenization to explicitly model structured periodicity (daily/weekly cycles), thereby mitigating modality mismatches and capturing multi-scale dependencies for improved long-term mobility prediction.

**Time Series Foundation Models.**  Existing time series foundation models can be divided into two categories: transformer-based models and language-based models. For transformer-based time series models [67, 39, 48], prior studies focus on transformer architecture and self-attention mechanisms to capture temporal dependency in time series data. For instance, PatchTST [48] introduces a patch-based self-attention mechanism to capture long-range dependencies in time series data. STanHop [67] and Crossformer [79] employ hierarchical self-attention to capture temporal dependencies and hierarchical structures in time series data. For language-based time series models [40, 32], prior studies adapt the LLMs to time series data and achieve state-of-the-art performance in time series forecasting tasks. For instance, AutoTime [40] introduces a novel autoregressive structure to capture the temporal dependency in time series data. Time-LLM [32] employs a large language model to capture the complex transitions of time series data. However, these models struggle to capture the inherent complexity of human mobility—with its abrupt location shifts and temporal dynamics—whereas RHYTHM leverages a novel spatio-temporal embedding paired with an autoregressive framework to effectively model these intricate transitions.

**Cross-domain Adaptation of LLMs.**  LLMs have evolved from specialized natural language processing systems into versatile foundation models capable of sophisticated reasoning across diverse tasks [1, 9]. Their transformer-based architecture and extensive pretraining have enabled remarkable transfer capabilities to domains beyond text. In computer vision, models like CLIP [53] align visual and textual representations for zero-shot recognition, while in time series analysis, approaches such as One-Fits-All [81] and LLM4TS [11] demonstrate competitive forecasting through tokenized numerical sequences. In biomedicine, BioBERT [35] and BioGPT [44] demonstrate significant gains on clinical NLP benchmarks, while instruction-tuned models like Med-PaLM approach expert-level medical QA performance [57]. In finance, domain-specific LLMs such as FinBERT [31] and BloombergGPT [70] substantially outperform general-purpose models on sentiment analysis and information extraction.

Rather than computationally expensive full fine-tuning, parameter-efficient adaptation strategies have gained prominence. Low-Rank Adaptation (LoRA) [29] introduces trainable low-rank matrices into attention layers, while more recent approaches keep LLMs entirely frozen by using lightweight adapters that project non-linguistic inputs into the model's embedding space. In vision, prefix-tuning methods [2, 60] train small encoders to produce "prompts" for frozen LLMs, while time series approaches [40, 32] employ projection layers to convert numeric sequences into token embeddings.

Applications of LLMs to human mobility modeling remain limited, with existing approaches relying primarily on parameter-intensive adaptation. Mobility-LLM [23] employs partial fine-tuning, while LLM-Mob [64] leverages in-context learning but lacks structured temporal modeling. In contrast, RHYTHM maintains a fully frozen LLM backbone, preserving the model's pre-trained knowledge while introducing a specialized spatio-temporal framework that efficiently adapts to mobility data characteristics.

## C  Attention Implementation Details

Our attention mechanism implements a pre-norm transformer block to enhance training stability, with a gated feed-forward network for improved expressivity. The mathematical formulation of our attention block is as follows:

$$Z = \text{LayerNorm}(X) + \text{Multi-Head Attention}(\text{LayerNorm}(X)),$$
$$\widetilde{Z} = Z + \text{GatedFFN}(\text{LayerNorm}(Z)),$$

where $X$ is the input sequence. The multi-head attention operation computes:

$$\text{Multi-Head}(X) = [\text{head}_1 \| \text{head}_2 \| \dots \| \text{head}_h] W_{\text{out}},$$
$$\text{head}_i = \text{Softmax}\left( \frac{X W_{q,i} (X W_{k,i})^\top}{\sqrt{d_k}} \right) X W_{v,i},$$

with $h$ attention heads, where $W_{q,i}, W_{k,i}, W_{v,i} \in \mathbb{R}^{d \times d_k}$ are the query, key, and value projection matrices for the $i$-th head, and $W_{\text{out}} \in \mathbb{R}^{d \times d}$ is the output projection matrix. The gated feed-forward network incorporates an adaptive gating mechanism:

$$\text{GatedFFN}(Z) = \text{FFN}(Z) \odot \sigma(W_{\text{gate}} Z),$$
$$\text{FFN}(Z) = W_2 \, \text{GELU}(W_1 Z),$$

where $\sigma$ denotes the sigmoid function, $\odot$ represents element-wise multiplication, and $W_{\text{gate}} \in \mathbb{R}^{d \times d}$ is the learnable gating matrix. The feed-forward network expands the hidden dimension by a factor of 4, with $W_1 \in \mathbb{R}^{4d \times d}$ and $W_2 \in \mathbb{R}^{d \times 4d}$. Dropout is applied after both the attention and feed-forward operations to prevent overfitting.

## D  Prompt Design Examples

---

**Trajectory Information**

```
This is the trajectory of user <User_ID> of day <Day_ID> which is a
<Day_of_Week>.  The trajectory consists of <N> records, each record of
coordinate is as follows:
08:30:  (X=136, Y=42);
09:00:  (X=136, Y=42);
09:30:  (X=137, Y=41);
10:00:  (X=146, Y=37);
10:30:  (X=145, Y=38);
11:00:  (X=144, Y=38);
11:30:  (X=135, Y=41);
12:00:  (X=135, Y=42);
12:30:  (X=135, Y=42);
13:00:  (X=135, Y=42).

Key transitions:  At 10:00:  (X=137, Y=41)→ (X=146, Y=37); At 11:30:  (X=144,
Y=38)  → (X=135, Y=41).

Main stay locations:  (X=136, Y=42) from 08:30 to 09:30 (0.5 hours); (X=145,
Y=38) from 10:00 to 11:00 (0.5 hours); (X=135, Y=42) from 11:30 to 13:00 (1.5
hours).
```

---

**Task Description**

```
You are a mobility prediction assistant that forecasts human movement
patterns in urban environments.  The city is represented as a 200 x 200
grid of cells, where each cell is identified by coordinates (X,Y). The X
```

```
coordinate increases from left (0) to right (199), and the Y coordinate
increases from top (0) to bottom (199).

TASK: Based on User <User_ID>'s historical movement patterns, predict their
locations for Day <Day_ID> (<Day_of_Week>).  The predictions should capture
expected locations at 30-minute intervals throughout the day (48 time slots).
The model should analyze patterns like frequent locations, typical daily
routines, and time-dependent behaviors to generate accurate predictions of
where this user is likely to be throughout the next day.

The previous days' trajectory data contains information about the user's
typical movement patterns, regular visited locations, transition times,
and duration of stays.  Key patterns to consider include:  home and work
locations, morning and evening routines, lunch-time behaviors, weekend vs.
weekday differences, and recurring visit patterns.
```

# E   Implementation Details

---

**Algorithm 1** RHYTHM – Overall Pipeline

---

**Require:** Trajectory $X = \{(t_i, l_i)\}_{i=1}^{T}$ (timestamps and location IDs); prediction horizon $\{t_{T+1}, \ldots, t_{T+H}\}$; segment length $L$; frozen LLM

1: **function** PRECOMPUTESEMANTICS($X, L, \{t_{T+1}, \ldots, t_{T+H}\}, \text{LLM}$)
2:  $N \leftarrow T/L$
3:  **for** $i = 1$ to $N$ **do**                 ▷ trajectory information
4:    $TE_i \leftarrow \text{SelectLast}(\text{LLM}(\text{Prompt}_{\text{seg}}(\{X_1, \ldots, X_T\})))$
5:  **end for**
6:  $TE^T \leftarrow \text{SelectLast}(\text{LLM}(\text{Prompt}_{\text{task}}(\{t_{T+1}, \ldots, t_{T+H}\})))$     ▷ task description
7:  **return** $\{TE_i\}_{i=1}^{N}$, $TE^T$
8: **end function**

9: **function** PREDICT($X, \{t_{T+1}, \ldots, t_{T+H}\}, \{TE_i\}_{i=1}^{N}, TE^T, L, \text{LLM}$)
10:  **for** $i = 1$ to $T$ **do**           ▷ embed time + location into token space
11:    $E_i^{\text{temporal}} \leftarrow \text{TemporalEmbed}(t_i)$
12:    $E_i^{\text{spatial}} \leftarrow \text{SpatialEmbed}(l_i)$
13:    $E_i \leftarrow E_i^{\text{temporal}} + E_i^{\text{spatial}}$
14:  **end for**
15:  Partition $\{E_i\}_{i=1}^{T}$ into $N = T/L$ segments $s_i = \{E_{(i-1)L+1 : iL}\}$
16:  **for** $i = 1$ to $N$ **do**          ▷ intra-segment attention, then pool to one token
17:    $\widetilde{E}^{(i)} \leftarrow \text{IntraAttention}(s_i)$
18:    $SE_i \leftarrow \text{Pool}(\widetilde{E}^{(i)})$
19:  **end for**
20:  $\{\widetilde{SE_i}\}_{i=1}^{N} \leftarrow \text{InterAttention}(\{SE_i\}_{i=1}^{N})$       ▷ model long-range dependency
21:  **for** $i = 1$ to $N$ **do**         ▷ additive alignment of semantics at segment level
22:    $CE_i \leftarrow \widetilde{SE_i} + TE_i$
23:  **end for**
24:  **for** $j = 1$ to $H$ **do**             ▷ future-time tokens + task semantics
25:    $E_{N+j} \leftarrow \text{TemporalEmbed}(t_{T+j})$
26:    $CE_{N+j} \leftarrow E_{N+j} + TE^T$
27:  **end for**
28:  $h_{1:N+H} \leftarrow \text{LLM}(\{CE_{1:N+H}\})$           ▷ frozen backbone forward
29:  $p_{1:H} \leftarrow \text{Softmax}(\text{ProjToClasses}(h_{N+1:N+H}))$
30:  **return** $\{\arg\max p_j\}_{j=1}^{H}$
31: **end function**

---

## F    Dataset

We provide the detail of datasets used in this paper as shown in Table 5.

Table 5: **Dataset Statistics**

| City | Users | Duration | Spatial Resolution | Places |
|------|-------|----------|--------------------|--------|
| Kumamoto | 3k | 75 days | 500m $\times$ 500m | 40k |
| Sapporo | 17k | 75 days | 500m $\times$ 500m | 40k |
| Hiroshima | 22k | 75 days | 500m $\times$ 500m | 40k |

## G    Experiment Settings

### G.1    Evaluation Metrics

**Accuracy@k** measures proportion of correct predictions within top-$k$ ranked locations:

$$\text{Accuracy@}k = \frac{1}{H}\sum_{i=1}^{H}\mathbb{1}(l_{T+i}\in \text{top-}k(\hat{p}_{T+i})),$$

where $\mathbb{1}(\cdot)$ is the indicator function and $\hat{p}_{T+i}$ is the predicted probability distribution.

**Mean Reciprocal Rank (MRR)** evaluates quality of ranked predictions:

$$\text{MRR} = \frac{1}{H}\sum_{i=1}^{H}\frac{1}{\text{rank}(l_{T+i})},$$

where $\text{rank}(l_{T+i})$ is the rank position of the true location.

**Dynamic Time Warping (DTW)** measures spatial similarity between trajectories:

$$\text{DTW}(\mathcal{Y},\hat{\mathcal{Y}}) = \min_{\pi}\sum_{(i,j)\in\pi}d(l_{T+i},\hat{l}_{T+j}),$$

where $\pi$ is a valid warping path and $d(\cdot,\cdot)$ is the Euclidean distance.

**BLEU** quantifies n-gram overlap between predicted and ground-truth sequences:

$$\text{BLEU} = BP\cdot\exp\left(\sum_{n=1}^{N}w_n\log p_n\right),$$

where $p_n$ is n-gram precision, $w_n$ is the weight for each n-gram level, and $BP$ is a brevity penalty.

### G.2    Computational Resource

We perform all experiments using a single NVIDIA A100 GPU with 40GB of memory and a 24-core Intel(R) Xeon(R) Gold 6338 CPU operating at 2.00GHz. Our code is developed in PyTorch [52] and utilizes the Hugging Face Transformer Library[2] for experimental execution.

### G.3    Hyperparameters

We present the hyperparameters used in the training stage for each model. Embeddings for time-of-day and day-of-week, the categorical location embedding, and the coordinate projection all use hidden dimensions of 128, 128, 256, and 128, respectively. We use **AdamW** [41] as the optimizer. For model training, we conduct a systematic hyperparameter search, exploring learning rates from the set $\{1\times10^{-4}, 3\times10^{-4}, 5\times10^{-4}\}$ and weight decay values from $\{0, 0.001, 0.01\}$. Through extensive validation experiments, we determine the optimal configuration for each dataset. All models are trained with a consistent batch size of 64 across all datasets for fair comparison. The final hyperparameter settings are selected based on performance on the validation set.

---

[2]https://huggingface.co/docs/transformers

### G.4 LLM variants

Our experiments deployed multiple foundation language models as text embedders and frozen back-bones within RHYTHM to evaluate cross-scale performance. Table 6 presents the pre-trained models accessed through the Hugging Face Transformers library, ranging from 125M to 3B parameters.

Table 6: **Pre-trained language models employed as backbones in RHYTHM.**

| Model | Parameters | HuggingFace Repository |
|---|---|---|
| OPT-125M | 125M | facebook/opt-125m |
| OPT-350M | 350M | facebook/opt-350m |
| Llama-3.2-1B | 1.24B | meta-llama/Llama-3.2-1B |
| Qwen-2.5-1.5B | 1.54B | Qwen/Qwen2.5-1.5B |
| DeepSeek-R1-1.5B | 1.78B | deepseek-ai/DeepSeek-R1-Distill-Qwen-1.5B |
| Gemma-2-2B | 2.61B | google/gemma-2-2b-it |
| Phi-2 | 2.78B | microsoft/phi-2 |
| Llama-3.2-3B | 3.21B | meta-llama/Llama-3.2-3B |

## H  Additional Experimental Results

### H.1  Autoregressive vs Non-autoregressive Strategy

Table 7 compares RHYTHM under autoregressive and non-autoregressive strategies. The non-autoregressive approach delivers comparable accuracy while being over two orders of magnitude faster.

Table 7: **Comparison of RHYTHM with autoregressive and non-autoregressive prediction strategies.** Results are reported using Acc@1, Acc@3, Acc@5, and computational time per iteration (s/iter). The best results are highlighted in **bold**.

| Model | Time (s/iter) | Acc@1 | Acc@3 | Acc@5 |
|---|---|---|---|---|
| Non-autoregressive | **0.96** | **0.2929** | 0.5200 | **0.5835** |
| Autoregressive | 39.80 | 0.2884 | **0.5247** | 0.5801 |

### H.2  Deployment efficiency compared to LLM-based baselines

Table 8 reports deployment efficiency for RHYTHM compared with LLM-based baselines, evaluated under both GPU and CPU inference. On GPU, RHYTHM requires substantially less memory and achieves lower latency than TimeLLM, while remaining competitive with Mobility-LLM. On CPU, RHYTHM also demonstrates favorable latency and moderate RAM usage, underscoring its practicality for deployment in resource-constrained environments.

Table 8: **Deployment efficiency of RHYTHM, TimeLLM, and Mobility-LLM. Deployment efficiency of RHYTHM, TimeLLM, and Mobility-LLM.** We report model size, GPU memory footprint (GRAM), inference latency on GPU and CPU, and RAM usage.

| Model | Size (MB) | GPU | | CPU | |
|---|---|---|---|---|---|
| | | GRAM (MB) | Latency (ms) | RAM (MB) | Latency (ms) |
| RHYTHM | 5841.9 | 5741.4 | $261.6 \pm 0.8$ | 12497.9 | $39.5 \pm 0.9$ |
| TimeLLM | 3762.7 | 11213.1 | $392.7 \pm 9.5$ | 18677.8 | $64.2 \pm 2.5$ |
| Mobility-LLM | 4880.7 | 9872.8 | $192.1 \pm 5.2$ | 10552.2 | $32.3 \pm 1.9$ |

# I  Resource Requirements and Computational Cost

In this section, we detail RHYTHM's resource requirements to provide a practical perspective on the computational cost of deploying RHYTHM.

## I.1  Dataset Preprocessing

Time and storage costs of semantic embedding generation are detailed in Table 9.

Table 9: **Preprocessing cost of RHYTHM across datasets.**

| Dataset | Preprocessing Time (h) | Storage Size (GB) |
|---|---|---|
| Kumamoto | 3.1 | 1.6 |
| Sapporo | 15.9 | 9.1 |
| Hiroshima | 20.1 | 11.8 |

## I.2  Training Resource Usage

Training requirements with varying sequence lengths are summarized in Table 10, including GPU memory (GRAM) and runtime per epoch. The results show that memory consumption and training time do not increase linearly with sequence length, demonstrating that RHYTHM can handle long sequences at moderate additional cost due to its temporal tokenization design.

Table 10: **Training cost with varying sequence lengths.**

| Sequence Length ($T$) | GRAM (MB) | Time/Epoch (min) |
|---|---|---|
| 48 | 7126.3 | 23.1 |
| 168 | 7469.5 | 24.6 |
| 336 | 8202.0 | 26.5 |
| 672 | 9826.4 | 30.9 |

# J  Additional Ablation Studies

## J.1  Segment Length Sensitivity

We analyze the impact of varying temporal segment length $L$ on prediction accuracy and efficiency in Table 11. Smaller segments (e.g., $L = 1$, 30 minutes) capture fine-grained details but result in excessive fragmentation and substantially higher computation time. In contrast, very large segments (e.g., $L = 96$, 2 days) reduce runtime but blur meaningful daily mobility boundaries, leading to degraded accuracy. Daily segmentation ($L = 48$) achieves the best balance, yielding the highest predictive performance while keeping iteration time under 1 second. These findings validate our choice of $L = 48$ as a balanced temporal unit for mobility modeling.

Table 11: **Effect of varying segment length $L$ on prediction accuracy and runtime.** The best results are highlighted in **bold**.

| Segment Length ($L$) | Segments ($N$) | Acc@1 | Acc@3 | Acc@5 | Time (s/iter) |
|---|---|---|---|---|---|
| 1 (30 min) | 336 | 0.2801 | 0.5049 | 0.5764 | 6.59 |
| 24 (12 hr) | 14 | 0.2883 | 0.5124 | 0.5792 | 1.10 |
| 48 (1 day) | 7 | **0.2929** | **0.5200** | **0.5835** | 0.96 |
| 96 (2 days) | 3 | 0.2851 | 0.5087 | 0.5743 | 0.93 |

## J.2 Impact of Pretrained LLMs

RHYTHM benefits substantially from pretraining: the pretrained LLM variant achieves the strongest accuracy, while randomly initialized and LLM-free variants lag behind as shown in Table 12.

Table 12: **Comparison of RHYTHM with pretrained, randomly initialized, and no-LLM variants on Kumamoto.** Pretraining provides the best accuracy across all metrics.

| RHYTHM Variant | Acc@1 | Acc@3 | Acc@5 |
|---|---|---|---|
| w/ pretrained LLM (ours) | **0.2929** | **0.5200** | **0.5835** |
| w/ randomly initialized LLM (frozen) | 0.2556 | 0.4842 | 0.5313 |
| w/o LLM (only attention) | 0.2749 | 0.4921 | 0.5623 |

## J.3 Computational Cost of RHYTHM Components

Table 13 shows how different architectural choices affect model size and training time. Temporal tokenization and hierarchical attention both contribute to reducing runtime without significantly increasing parameter count, highlighting their role in RHYTHM's efficient design.

Table 13: **Trainable parameters (absolute and relative) and training time per epoch for different architectural configurations of RHYTHM on Kumamoto.**

| Configuration | Trainable Params (MB) | Share of Total (%) | Time/Epoch (min) |
|---|---|---|---|
| Full RHYTHM (frozen LLM) | 152 | 12.37 | 31 |
| Unfrozen LLM w/ LoRA | 200 | 16.27 | 102 |
| w/o Temporal Tokenization | 148 | 12.00 | 53 |
| Only Attention Module (no LLM) | 152 | 12.37 | 16 |
| w/o HA | 39 | 3.25 | 20 |

