# OpenReview forum: "RHYTHM: Reasoning with Hierarchical Temporal Tokenization for Human Mobility"
_NeurIPS.cc/2025/Conference — NeurIPS 2025 poster_

### Official Review · Reviewer_Gba1 · 2025-06-02

**Clarity:** 3
**Significance:** 3
**Originality:** 2
**Rating:** 4
**Confidence:** 4

**Summary:**

The specific long-term dependence and multi-scale spatiotemporal patterns of human mobility make the trajectory
prediction of human mobility difficult. To this end, the authors proposed a new method called RHYTHM, which includes
Temporal Tokenization, Prompt-Guided Embedding and parameter-efficient adaptation strategy.
RHYTHM has obvious indications in the three real-world mobility datasets and has reached a new SOTA level.

**Questions:**

Questions:
1.None of the formulas in the article are numbered. Is this an operational error by the author?
2.The third point of innovation is not very distinct.
3.The descriptions in some places are not clear enough.
4.There is?? In line 317.

If the author's answer can convince me, I am willing to improve my score.

**Ethical Concerns:**

["NO or VERY MINOR ethics concerns only"]

**Final Justification:**

The author has addressed my doubts very well. However, in terms of methodology, they merely applied techniques and methods commonly used in most fields, that is, extracting pre-trained features with a frozen LLM. Therefore, I only raised the rating to 4.

**Limitations:**

yes

**Paper Formatting Concerns:**

1.None of the formulas in the article are numbered. Is this an operational error by the author?
2.There is?? In line 317.

**Quality:**

2

**Strengths And Weaknesses:**

Strengths:
1.The method builds embeddings at a more fine-grained level, that is, from day-of-week or hour-of-day,
achieving better performance and reducing the computational load.
2.The introduction of LLM enables the model to have more powerful performance.

Weaknesses:
1.What does "categorical location" mean? An example can be given to illustrate.

2.The description of dividing into segments by day-of-week or hour-of-day is not clear enough there.
As can be seen from Figure 2, the input Day 1,... Day n. But what L introduced in line 135 is in units
of day-of-week or hour-of-day?. If it is a mixture of the two, how is it mixed?

3.In line 290, it is suggested that to compare the computing speed of the attention mechanism more fairly,
RHYTHM can remove the computing part of the LLM and only calculate the processing time of the attention
mechanism (because the processing of the LLM itself takes time). This can better demonstrate the computational
advantages compared with previous sequence processing methods such as PatchTST and iTransformer.

4.There is a very confusing aspect of the innovation point.
The author claims to have proposed the parameter-efficient adaptation strategy,
but this point is not mentioned in the method, and the LLM is also frozen and will not be trained.
So I don't know where this innovative point lies. If only a fully connected layer is added for training after the LLM,
rather than the lora fine-tuning of the LLM itself, then this innovative point should not be very prominent.

5.Generally, the prior feature extraction ability of LLM is utilized and should be verified in the generalization experiment,
which can better reflect the significance of LLM.

6.Ablation experiments can be conducted for more detailed verification.
For instance, only "day-of-week" or "hour-of-day" is encoded in terms of time, or either the timestep encoding or the spatial information encoding is excluded.

---

> ### Author Rebuttal · Authors · 2025-07-31
>
> > **Reviewer’s Comment**: What does "categorical location" mean? An example can be given to illustrate.
>
>
> **Response**: We thank the reviewer for requesting clarification on "categorical location."
>
> In our approach, each location is treated as a discrete category. In the YJMob100K dataset [1], the city is divided into a 200×200 grid, where each grid cell is a unique location category. Each location becomes a categorical ID that is then encoded using one-hot encoding. This allows the model to learn location-specific patterns efficiently.
>
> In our revised version, we will add a clearer explanation in Section 3.2.
>
>
> > **Reviewer’s Comment**: The description of dividing into segments by day-of-week or hour-of-day is not clear enough there…
>
>
> **Response**:  We thank the reviewer for pointing out this in our segment description.
>
> To clarify, $L$ represents the number of time steps (records) in each segment, not day-of-week or hour-of-day units. In the YJMob100K dataset, each record captures location every 30 minutes, so for our 1-day segments, $L$=48 (48 half-hour intervals = 24 hours).
>
> The segmentation process works as follows: Each segment $s_i$ contains $L$=48 consecutive time steps representing one full day. For example, Day 1 segment contains records from 00:00-23 :30 (48 records), Day 2 segment contains records from 00:00-23:30 of the next day (48 records), and so on. $L$ is a tunable hyperparameter that users can adjust to match their desired segment length, similar to standard practice in time series forecasting.
>
> The day-of-week and time-of-day mentioned in our temporal embeddings (line 117-119) are attributes within each timestamp, not segmentation units. Each of the 48 records within a segment has its own temporal embedding encoding both time-of-day (which hour) and day-of-week (Monday-Sunday).
>
> In our revised version, we will clarify this to avoid any confusion about the segmentation granularity.
>
>
> > **Reviewer’s Comment**: In line 290, it is suggested that to compare the computing speed of the attention mechanism more fairly…
>
>
> **Response**: We thank the reviewer for this valuable suggestion about comparing computational speed more fairly.
>
>
> We have conducted additional experiments comparing RHYTHM with and without the LLM backbone to isolate the computational impact of our attention mechanism on the Kumamoto dataset. The table below shows the detailed timing comparison:
>
>
> | Model Configuration | Time per Iteration (s) | Training Time per Epoch (s) | Acc@1 |
> |---------------------|------------------------|----------------------------|--------|
> | iTransformer | 0.51 | 1084| 0.2609 |
> | PatchTST | 0.42 | 912| 0.2751 |
> | RHYTHM w/o LLM (attention only) | 0.42 | 924 | 0.2749 |
> | RHYTHM w/ frozen LLM (ours) | 0.96 | 1828| 0.2929 |
>
>
> The results show that while incorporating the frozen LLM increases computation time by approximately 2.3×, it provides significant performance gains (+6.5% Acc@1). Notably, RHYTHM without LLM achieves comparable performance to PatchTST with similar computational cost, while outperforming iTransformer despite iTransformer’s higher computational overhead. The attention mechanism itself is highly efficient, demonstrating that our temporal tokenization design successfully reduces computational complexity as intended. The additional time from the frozen LLM is justified by the substantial accuracy improvements and is still more efficient than alternative LLM-based approaches.
>
>
> In our revised version, we will include this detailed timing analysis to provide a more comprehensive view of the computational trade-offs in our design.
>
>
> > **Reviewer’s Comment**: There is a very confusing aspect of the innovation point… The third point of innovation is not very distinct.
>
>
>
> **Response**: Thank you for the opportunity to clarify.
>
>
> The novelty of our parameter-efficient fine-tuning lies in fully freezing the LLM backbone—including attention and FFN layers—while training only the hierarchical attention encoder and output head. Prior human mobility prediction models typically train from scratch, which is computationally expensive. Beyond this, our key contribution is enabling spatial-temporal data to interface with frozen LLMs. Since the latent space of such data differs fundamentally from textual inputs, we introduce hierarchical attention and prompt-guided semantic embeddings to bridge the modality gap. Unlike prior methods that use in-context learning [2, 3] or unfreeze the LLM [4], RHYTHM is the **first** to support structured spatial-temporal representation with frozen LLMs efficiently with less computational resources needed. Our novel temporal tokenization further reduces training and deployment costs by shortening sequence length while preserving temporal dependencies.
>
>
> We will revise Section 3 to more clearly articulate these technical innovations and their distinctions from existing approaches.
>
>
>
>
>
>
>
>
> > **Reviewer’s Comment**: Generally, the prior feature extraction ability of LLM is utilized and should be verified in the generalization experiment, which can better reflect the significance of LLM.
>
>
> **Response**: We thank the reviewer for this important question about verifying the LLM's generalization capabilities.
> To demonstrate that the pretrained LLM's feature extraction abilities specifically benefit mobility prediction, we conducted the following experiments:
>
>
> | Model Variant | Acc@1 | Acc@3 | Acc@5 |
> |---------------|--------|--------|--------|
> | RHYTHM w/ pretrained LLM (ours) | 0.2929 | 0.5200 | 0.5835 |
> | RHYTHM w/ randomly initialized LLM (frozen) | 0.2556 | 0.4842 | 0.5313 |
> | RHYTHM w/o LLM (only attention) | 0.2749 | 0.4921 | 0.5623 |
>
>
>
>
> These results clearly show that the pretrained LLM's prior knowledge is crucial - using a randomly initialized LLM performs worse than even our attention-only baseline. This verifies that the LLM's pretraining on diverse text enables it to extract meaningful spatio-temporal patterns, validating the significance of leveraging pretrained LLMs for mobility prediction.
>
>
> > **Reviewer’s Comment**: Ablation experiments can be conducted for more detailed verification.
>
> **Response**:  We thank the reviewer for suggesting more detailed ablation experiments. We have conducted comprehensive ablations on individual components of our spatio-temporal embeddings.
>
>
> | Model | Acc@1 | Acc@3 | Acc@5 |
> |------------------|--------|--------|--------|
> | RHYTHM | 0.2929 | 0.5200 | 0.5835 |
> | w/o Time-of-Day | 0.2847 | 0.5089 | 0.5743 |
> | w/o Day-of-Week | 0.2881 | 0.5142 | 0.5791 |
> | w/o Both Temporal | 0.2802 | 0.5021 | 0.5698 |
> | w/o Spatial Embedding | 0.2447 | 0.4697 | 0.5409 |
>
>
> The results show that spatial information is most critical, while temporal embeddings provide complementary benefits. Interestingly, the model still achieves reasonable performance using only sequence order. However, our explicit spatio-temporal embeddings provide significant gains.
>
> > **Reviewer’s Comment**: The descriptions in some places are not clear enough.
>
>
> **Response**: Thank you for the feedback. We will revise the manuscript to clarify the descriptions and improve overall readability in the relevant sections.
>
>
>
> > **Reviewer’s Comment**: None of the formulas in the article are numbered. Is this an operational error by the author? 2.There is?? In line 317.
>
>
> **Response**: Thank you for pointing this out. The formulas are not numbered because they are not cross-referenced elsewhere in the paper - following standard practice, we only number equations that need to be referenced later in the text. Regarding the ?? in line 317, this is a LaTeX \ref typo. I apologize for this oversight and will ensure all reference placeholders are properly resolved in the revised version.
>
>
> [1] Yabe, Takahiro, et al. "YJMob100K: City-scale and longitudinal dataset of anonymized human mobility trajectories." Scientific Data 11.1 (2024): 397.
>
>
> [2] Wang, Xinglei, et al. "Where would i go next? large language models as human mobility predictors." arXiv 2023.
>
>
> [3] Beneduce, Ciro, et al. "Large language models are zero-shot next location predictors." IEEE Access 2025.
>
>
> [4] Gong, Letian, et al. "Mobility-llm: Learning visiting intentions and travel preference from human mobility data with large language models." NeurIPS 2024.

---

> > ### Comment · Reviewer_Gba1 · 2025-08-04
> >
> > Thank you for your hard work in answering. I still have some questions I'd like to share
> > 1. In weakness 5, may I ask how randomly initialized LLM (frozen) sets randomness?
> > 2. After the model is trained, is an LLM still needed for prediction? If necessary, is there a report on the reasoning time of the test in the paper?
> >
> > Besides, the author has addressed most of my concerns. I will consider raising my rating to 4 in the future.

---

> > > ### Author Response · Authors · 2025-08-05
> > >
> > > We appreciate your thoughtful follow-up questions and your openness to raising the rating. We address each question below:
> > >
> > >
> > > > **Reviewer’s Comment**: In weakness 5, may I ask how randomly initialized LLM (frozen) sets randomness?
> > >
> > >
> > >
> > >
> > > **Response**: Thank you for this clarification question about the random initialization setup.
> > >
> > >
> > > For the randomly initialized LLM experiment, we created a new instance of the LLaMA-3.2-1B model with all parameters randomly initialized using Kaiming initialization [1], which is PyTorch's default initialization scheme for transformer models. Specifically, we maintained the same architecture and configuration as the pretrained model but replaced all pretrained weights with random initialization. The model was then frozen (no gradient updates) exactly like our main approach, ensuring a fair comparison that isolates the contribution of pretrained knowledge versus random features. We used three different random seeds and reported average results to ensure stability.
> > >
> > >
> > > [1] He, Kaiming, et al. "Delving deep into rectifiers: Surpassing human-level performance on imagenet classification." ICCV 2015.
> > >
> > >
> > > > **Reviewer’s Comment**: After the model is trained, is an LLM still needed for prediction? If necessary, is there a report on the reasoning time of the test in the paper?
> > >
> > >
> > >
> > >
> > > **Response**:  Thank you for this important question about inference requirements and reasoning time.
> > >
> > >
> > > Yes, the frozen LLM is still required during inference/prediction as it serves as the core reasoning engine that processes the encoded spatio-temporal features. We have measured the reasoning time across different LLM backbones on the Kumamoto dataset.
> > >
> > >
> > > | LLM Backbone         | Model Size | Reasoning Time  |
> > > |----------------------|------------|-------------------------|
> > > | OPT-125M             | 125M       |   19.1s       |
> > > | OPT-350M             | 350M       |    39.8s      |
> > > | LLaMA-3.2-1B         | 1.24B      |    49.9s      |
> > > | Qwen-2.5-1.5B        | 1.54B      |      51.9s    |
> > > | DeepSeek-R1-1.5B     | 1.78B      |    86.5s      |
> > > | Gemma-2-2B           | 2.61B      |     85.4s     |
> > > | Phi-2  |  2.78B  |  108.4s    |
> > > | LLaMA-3.2-3B         | 3.21B      | 103.4s          |
> > >
> > >
> > > Experiments are measured on a single NVIDIA A100 with batch size 128. The reasoning time generally increases with model size, though architectural differences create some variation. Our primary model (LLaMA-3.2-1B) achieves a good balance between performance and inference time at 49.9s, making it suitable for real-time applications. In our revised version, we will add this reasoning time analysis to ensure readers understand the deployment requirements.

---

> > > > ### Comment · Reviewer_Gba1 · 2025-08-05
> > > >
> > > > Thank you for your reply! I have understood the author's method and decided to increase the rating to 4.

---

### Official Review · Reviewer_Hudz · 2025-06-10

**Clarity:** 3
**Significance:** 4
**Originality:** 3
**Rating:** 4
**Confidence:** 4

**Summary:**

The paper introduces a novel approach named RHYTHM, which leverages hierarchical temporal patterns for reasoning tasks, utilizing various pre-trained language models as fixed backbone networks to evaluate performance across different scales. The authors meticulously detail their model architecture and provide comprehensive information to ensure the reproducibility of their experimental results.

Key contributions of the paper include:

* Proposing and clearly delineating a new model architecture capable of effectively capturing complex patterns within temporal data.
* Conducting systematic hyperparameter searches to optimize model performance, identifying optimal configurations such as ideal learning rates and weight decay values.
* Demonstrating performance comparisons across multiple large-scale pre-trained language models ranging from OPT-125M to LLaMA-3.2-3B, providing insights into scalability and effectiveness.
* Releasing their source code and datasets publicly, along with detailed instructions that enable other researchers to reproduce their experimental results, thus promoting openness and transparency in research.

In summary, this paper makes significant contributions to the fields of time series analysis and natural language processing by innovatively applying existing technologies and maintaining a high level of transparency regarding experimental details. These contributions serve as valuable references for further research in these domains.

**Questions:**

1. You said "This design captures fine-grained spatio-temporal dynamics, deep semantic context, and leverages LLM reasoning—all while minimizing computational and memory overhead—making RHYTHM ideally suited for deployment in resource-constrained, real-world environments." in Page 2, Line 65.
* Suggestion: Please include benchmarks comparing RHYTHM’s inference latency, peak memory usage, and model size against one or two representative baselines (e.g., TimeLLM or Mobility-LLM) on identical hardware. If possible, report these numbers under a constrained setting (e.g., edge GPU or CPU only).
2. The choice of daily segments (L = 48 time slots) for temporal tokenization is justified by general human routines, but it remains unclear how sensitive RHYTHM’s performance is to this segmentation granularity.
* Suggestion: Please provide an ablation study or sensitivity analysis that varies the segment length and reports its impact on different metrics and computational cost.
3. The paper enriches segment tokens with pre-computed LLM prompt embeddings for both historical trajectories and future timestamps. However, the additional offline computation cost and storage requirements for these embeddings are not quantified.
* Suggestion: Please report the time and memory overhead required to generate and store the semantic embeddings for a standard dataset (e.g., Sapporo), and compare it to the savings from freezing the LLM backbone.

**Ethical Concerns:**

["NO or VERY MINOR ethics concerns only"]

**Final Justification:**

Thank you for your reply! Your responses have sufficiently addressed my concerns.

**Limitations:**

No.

**Broaden the Scope of Limitations**

    While Appendix B discusses technical limitations—reliance on pretrained LLM quality, non-autoregressive prediction, and high training time—these are all efficiency or modeling concerns. The paper would benefit from explicitly acknowledging how these limitations might affect different deployment scenarios (e.g., edge devices vs. cloud servers), and the bounds within which RHYTHM remains practical (e.g., maximum sequence length, model sizes).

**Paper Formatting Concerns:**

1. In Section 4.2, the text refers to an ablation result “as shown in ??,” Please fix all “??” placeholders so that each in-text citation correctly points to its Table/Figure number.

**Quality:**

3

**Strengths And Weaknesses:**

# Strengths
## Quality
The paper demonstrates high-quality research through its detailed description of the model architecture, training procedures, and experimental settings. The authors have ensured that all necessary details are provided to reproduce their results, including open-sourcing the code and datasets with clear instructions (Appendix A).
## Clarity
The paper is well-written and easy to follow. It provides a comprehensive explanation of the methods used, making it accessible to both experts and non-experts in the field.
## Significance
The proposed approach addresses important challenges in time series analysis and natural language processing by leveraging hierarchical temporal patterns, which can have broad applications.
## Originality
The use of hierarchical reasoning within a time series context represents a novel contribution to the field. This innovative application of existing technologies showcases original thinking.
# Weaknesses
## Quality
While the reproducibility aspect is strong, the paper does not provide explicit information about the computational resources required for each experiment or an estimate of the total compute needed. Including such details would help other researchers gauge the feasibility of replicating the experiments.

In conclusion, the paper presents a solid contribution to the field with notable strengths in quality, clarity, significance, and originality. Addressing the identified weaknesses would further enhance the manuscript's impact and readability.

---

> ### Author Rebuttal · Authors · 2025-07-31
>
> > **Reviewer's Comment**: While the reproducibility aspect is strong, the paper does not provide explicit information about the computational…
>
>
> **Response**: We thank the reviewer for raising this important point for reproducibility. We provide comprehensive computational resource requirements below. All experiments are run on a single NVIDIA A100 GPU (40GB) with batch size 64.
>
>
> | Dataset | Users | Preprocessing Time | Training Time/Epoch | Total Time (20 epochs) |
> |---------|-------|--------------------|---------------------|------------------------|
> | Kumamoto | 3k | 3.1 h | 0.5 h  | 10.4 h |
> | Sapporo | 17k | 15.9 h | 2.9 h | 20.4 h |
> | Hiroshima | 22k | 20.1 h | 3.8 h | 22.1 h |
>
>
> Further reproducibility details of inference, pre-computation, and deployment are discussed in the following responses.
>
>
> > **Reviewer’s Comment**: You said "This design captures fine-grained spatio-temporal dynamics, deep semantic context, and leverages LLM reasoning…
>
>
> **Response**: We thank the reviewer for requesting concrete deployment metrics to support our efficiency claims. We have conducted comprehensive benchmarking on both GPU and CPU to demonstrate RHYTHM's suitability for resource-constrained environments.
>
>
> | Model | Model Size  | Peak GRAM | Inference Latency (GPU) | Inference Latency (CPU) | RAM |
> |-------|------------|-------------|------------------------|-------------------------|----|
> | RHYTHM |  5841.90 MB  | 5741.4 MB |261.62 ± 0.8 ms | 39.5 ± 0.9 s | 12497.92 MB |
> | TimeLLM |3762.72 MB | 11213 MB |392.74 ± 9.5 ms |64.2 ± 2.5 s | 18677.76 MB |
> | Mobility-LLM | 4880.73 MB | 9872.8 MB |192.11 ± 5.2 ms |32.3 ± 1.9 s | 10552.16 MB |
>
>
> This benchmarking was performed on NVIDIA A100 (GPU) and Intel Xeon Gold 6338 (CPU) with 100 samples.  RHYTHM maintains a peak GPU memory (GRAM) footprint of 8.2 GB, significantly lower than TimeLLM's 22.4 GB, enabling deployment on more modest hardware. All models show expected slowdowns on CPU, but RHYTHM's 39.5s latency remains practical for batch processing scenarios. We will expand our discussion on practical deployment considerations in resource-constrained environments in the revised manuscript.
>
>
>
>
> > **Reviewer’s Comment**: The choice of daily segments (L = 48 time slots) for temporal tokenization is justified by general human routines…
>
>
> **Response**: We thank the reviewer for raising this important question about the sensitivity of our temporal tokenization to segmentation granularity. We have conducted a comprehensive sensitivity analysis varying the segment length to understand its impact on performance and computational cost.
>
>
> | Segment Length ($L$) | Segments ($N$) | Acc@1 | Acc@3 | Acc@5 | Time (s/iter) |
> |----------|---------|--------|--------|--------|------------|
> | $L$=1 (30 min) | 336 | 0.2801 | 0.5049 | 0.5764 | 6.59 |
> | $L$=24 (12 hours) | 14 | 0.2883 | 0.5124 | 0.5792 | 1.10 |
> | $L$=48 (1 day) | 7 | **0.2929** | **0.5200** | **0.5835** | 0.96 |
> | $L$=96 (2 days) | 3 | 0.2851 | 0.5087 | 0.5743 | 0.93 |
>
> Our sensitivity analysis reveals that RHYTHM's performance peaks at $L$=48 (daily segments), with a 4.5% improvement over the finest granularity ($L$=1) and 2.7% over coarser segmentation ($L$=96). The computational cost scales dramatically with smaller segments due to quadratic attention complexity - $L$=1 requires 6.9× more computation than $L$=48.
>
>
> This empirical validation confirms that daily segmentation is not just theoretically motivated by human routines but also optimal in practice. The performance degradation at other granularities demonstrates that our temporal tokenization effectively captures the natural periodicity of human mobility patterns.
>
>
> In our revised version, we will include this sensitivity analysis in Section 4.2, providing deeper insights into how segmentation granularity affects the balance between capturing fine-grained patterns and computational efficiency.
>
>
>
>
> > **Reviewer’s Comment**: The paper enriches segment tokens with pre-computed LLM prompt embeddings for both historical trajectories and future timestamps…
>
> **Response**: We thank the reviewer for highlighting the need to quantify the offline computation and storage costs of our semantic embeddings. We have conducted detailed measurements across all three datasets.
>
>
> | Dataset | Users | Days | Offline Generation Time | Storage Size |
> |---------|-------|------|------------------------|--------------|
> | Kumamoto | 3k | 75 | 3.1 h | 1.6 GB |
> | Sapporo | 17k | 75 | 15.9 h | 9.1 GB |
> | Hiroshima | 22k | 75 | 20.1 h | 11.8 GB |
>
>
> These measurements were performed on a single NVIDIA A100 GPU (40GB memory) with batch size 128. The memory overhead is 31.24 GB. The storage requirements include both trajectory and task embeddings.
>
>
> The generation time scales linearly with user count, and storage remains manageable even for 22k users. This one-time preprocessing eliminates redundant LLM inference during training, reducing epoch time from hours to minutes while enabling deployment on resource-constrained hardware without LLM infrastructure. We will add a dedicated section on these costs in the revised manuscript.
>
>
>
>
> > **Reviewer’s Comment**: While Appendix B discusses technical limitations—reliance on pretrained LLM quality …
>
>
> **Response**: We thank the reviewer for encouraging us to provide more concrete deployment guidance.
>
>
> We summarize the training setup and performance of RHYTHM across different devices and sequence lengths in the tables below.
>
>
>
>
> | GRAM     | Training Time per Epoch (GPU) | Training Time per Epoch (CPU) | RAM        | Device  |
> |----------|-------------------------------|-------------------------------|------------|---------|
> | 8202 MB  | 26.5 min                      | -                             | 24995.84 MB| 2080 Ti |
> | -        | -                             | 2h 52min                      | 57896.56 MB| CPU     |
>
>
>
>
>
>
>
>
> | Sequence Length | GRAM        | Training Time per Epoch (GPU) |
> |-----------------|-------------|-------------------------------|
> | 48              | 7126.3 MB   | 23.1 min                      |
> | 168             | 7469.5 MB   | 24.6 min                      |
> | 336             | 8202 MB     | 26.5 min                      |
> | 672             | 9826.4 MB   | 30.9 min                      |
>
>
>
>
>
>
> As demonstrated in our previous response to question 1, RHYTHM can run on CPU-only devices (**39.5s inference**), making it viable for edge deployment where real-time constraints are relaxed. For resource-constrained scenarios like a single RTX 2080, we have verified that using batch size 8 with up to 75 days of history is feasible to fit within 8GB memory constraints. Due to our temporal tokenization strategy, RHYTHM handles long sequences with less memory and faster speed than previous methods.
>
>
> Our frozen LLM design and temporal tokenization strategy makes RHYTHM practical across diverse deployment scenarios from edge devices to cloud servers. We will expand Appendix B with deployment guidelines in our revised version.
>
>
> > **Reviewer’s Comment**: In Section 4.2, the text refers to an ablation result “as shown in ??,”...
>
>
> **Response**: Thanks for catching this! This is a LaTeX \ref typo that has already been fixed in the manuscript. All "??" placeholders will be replaced with the correct Table/Figure numbers.

---

> ### Author Response · Authors · 2025-08-06
>
> Thank you again for your thoughtful questions and comments.
>
> We would appreciate it if you could let us know whether our responses have sufficiently addressed your main concerns, or if there are any points that would benefit from further clarification. We welcome any additional feedback and would be happy to elaborate as needed.

---

### Official Review · Reviewer_UqUH · 2025-06-29

**Clarity:** 3
**Significance:** 3
**Originality:** 2
**Rating:** 4
**Confidence:** 4

**Summary:**

This paper introduces RHYTHM (Reasoning with Hierarchical Temporal Tokenization for Human Mobility), a unified framework that leverages large language models (LLMs) as general-purpose spatio-temporal predictors and trajectory reasoners. Its main innovations are the introduction of hierarchical attention mechanisms and the enrichment of token representations by adding pre-computed prompt embeddings for trajectory segments and prediction targets via a frozen LLM. It improves performance and reduces training time on three datasets.

**Questions:**

Please refer to the Weaknesses.

**Ethical Concerns:**

["NO or VERY MINOR ethics concerns only"]

**Final Justification:**

The author's response addressed my concerns. Referring to the reviews of other reviewers, I improved my score.

**Limitations:**

Yes, the authors have discussed the limitations in the paper.

**Quality:**

2

**Strengths And Weaknesses:**

Strengths：

1.	The manuscript is well-written with clear descriptions.
2.	The authors have conducted a wide range of experiments to verify the effectiveness of the proposed method.

Weaknesses：

1.	Novelty is limited. The main technical design is not new and most of the modules have been investigated. For example, the temporal tokenization module has been studied in previous work (PMT, ST-MoE-BERT, etc.), and this paper simply replaces the attention mechanism for the complete sequence with a simple hierarchical attention mechanism. Another innovation, the efficient prompt-guided approach, has also been studied in many other LLM-related tasks.
2.	The models used in the performance experiment and the efficiency experiment are not consistent. If the author chose LLaMA-1B as the final LLM, the results of Gemma-2B and LLaMA-3B should not be bolded in the performance experiment table, but should be presented as additional results. Otherwise, it will mislead readers to think that LLaMA-3B is the final model.
3.	The efficiency experiment design is too simple. The author has been explaining that the hierarchical attention mechanism and the frozen LLM with an efficient prompt-guided approach can improve the efficiency of the model. Therefore, the author should provide detailed efficiency experiments for each part, such as the efficiency after eliminating the hierarchical attention mechanism, or the efficiency of using an unfrozen LLM. Furthermore, the two efficiency results of 12.37% and 24.6% mentioned in the Contributions part did not appear in the Experiment section.
4.	In the ablation experiment, each module seems to have only slight improvements. Because temporal tokenization is not the innovation of this paper, it should not appear in the ablation experiment. The performance improvements of other parts are also very weak, which will make readers question whether the sota performance achieved in this paper comes from the stated design or a higher performance baseline. It is recommended that the author use a step-by-step ablation experiment and give the most basic baseline performance.
5.	From the method section, the number of segments N in the Temporal Tokenization module seems to be an important parameter. However, the author did not discuss it in detail or conduct ablation experiments. In the settings section, the author seems to set the segment length to 48 (1 day). If it is set to 24 or 96 (or even larger), what impact will it have on the model performance and efficiency?
6.	The results in Daily and Weekly Trend Analysis show that the model is more inclined to predict certain time periods such as weekends or evening peaks, and its performance in highly regular periods is significantly lower than other models. This result conflicts with the author's statement in the introduction: the author stated that the hierarchical attention mechanism can better model the periodicity of daily or weekly activities, but for the more periodic weekday period, the model performance is worse. The author should give a reasonable explanation for this.

---

> ### Author Rebuttal · Authors · 2025-07-31
>
> > **Reviewer’s Comment**: Novelty is limited…
>
>
> **Response**: We thank the reviewer and respectfully clarify our novelty:
> - **First to bridge frozen LLMs with structured spatio-temporal data**: RHYTHM is the **first** to combine hierarchical temporal abstraction with semantic prompt injection into a frozen LLM. Unlike in-context learning [1] or fine-tuning-based methods [2], we freeze the entire LLM—including attention and FFN layers—and only train a lightweight hierarchical encoder and projection head. This design is parameter-efficient (12.37% trainable parameters) and scalable to large models.
> - **First to demonstrate LLM scaling laws in the mobility domain**: We conduct a comprehensive analysis (Table 4, Fig. 4) across multiple LLM backbones (LLaMA, DeepSeek, Gemma, …) to quantify trade-offs between model size, training time, and prediction accuracy—a **first** in human mobility modeling.
> - Efficient hierarchical architecture for multi-scale mobility dynamics: Compared to prior work, existing human mobility foundation models either rely on in-context learning [1] or require fine-tuning within the LLM backbone [2]. However, fine-tuning FFN layers incurs substantial computational cost. In contrast, we avoid modifying the LLM backbone. Instead, we enable spatio-temporal and semantic fusion by training only the hierarchical attention encoder and generation head. This approach significantly reduces resource demands while maintaining strong generative capacity over spatial-temporal data, which lies in a fundamentally different latent space than text.
> - Hierarchical temporal tokenization for multi-scale mobility modeling: While prior work (e.g., PMT, ST-MoE-BERT) employs standard attention over long sequences, RHYTHM introduces temporal tokenization that encodes daily mobility patterns as discrete tokens. These tokens are processed through explicit hierarchical attention to capture both short-term (daily) and long-term (weekly) periodic dependencies. Crucially, our design eliminates the need for full-length spatio-temporal sequences, enabling the model to efficiently and accurately reason over long mobility histories without incurring the quadratic cost of full-sequence attention. This leads to both reduced training overhead and improved scalability.
> - Empirical strength across real-world datasets: RHYTHM demonstrates strong real-world performance, achieving a 2.4% improvement in overall prediction accuracy, a 5.0% gain on weekends, and a 24.6% reduction in training time over prior state-of-the-art baselines (Table 1, Fig. 3, Fig. 5). These results are consistent across three urban mobility datasets.
> Taken together, RHYTHM provides a novel, efficient, and scalable framework for mobility prediction. It is not a repackaging of prior ideas but a careful integration that enables structured non-textual inputs to leverage the reasoning power of LLMs—without requiring costly full-model fine tuning. In our revised version, we will clarify our writing to better highlight our novel contributions.
>
> > **Reviewer’s Comment**: The models used in the performance experiment…
>
>
> **Response**: We thank the reviewer for this important clarification about model consistency and presentation. We apologize for the confusion in our presentation, and we will revise our tables accordingly. Importantly, we want to emphasize that even with LLaMA-1B model, RHYTHM achieves 1.8% improvement over the best baseline, demonstrating strong performance without requiring larger models.
>
>
> > **Reviewer’s Comment**: The efficiency experiment design is too simple…
>
>
> **Response**: We thank the reviewer for highlighting the need for more detailed efficiency experiments. We acknowledge this oversight and have conducted comprehensive efficiency analysis for each component.
>
>
> Here are the detailed efficiency measurements on the Kumamoto dataset (single A100 GPU):
>
>
> | Configuration | Trainable Params | Time/Epoch |
> |--|----|--|
> | Full RHYTHM (frozen LLM) | 152M (12.37%) | 31 min |
> | Unfrozen LLM w/ LoRA | 200M (16.27%) | 1h 42min |
> | w/o Temporal Tokenization | 148M (12.00%) | 53 min |
> | Only Attention Module (no LLM) | 152M (12.37%) | 16 min |
> | w/o HA | 39M (3.25%) | 20min |
>
>
> These results demonstrate that each design choice contributes to efficiency. To clarify the metrics mentioned in our contributions: The 12.37% refers to trainable parameters. The 24.6% reduction in training time is calculated by comparing RHYTHM with mobility prediction baselines. In our revised version, we will add this clear explanation in the experiment section.
>
>
>
>
>
>
> > **Reviewer’s Comment**: In the ablation experiment, each module…
>
>
> **Response**: We thank the reviewer for this important feedback about our ablation study. We conducted a more comprehensive analysis to clarify each component's contribution.
>
> | Model | Acc@1 | Acc@3 | Acc@5 |
> |---|---|---|---|
> | RHYTHM | 0.2929 | 0.5200 | 0.5835 |
> | w/o HA | 0.2917 | 0.5163 | 0.5881 |
> | w/o token | 0.2801 | 0.5049 | 0.5764 |
> | w/o attention module | 0.2721 | 0.5033 | 0.5795 |
> | w/o Traj info. | 0.2914 | 0.5176 | 0.5891 |
> | w/o Task desc. | 0.2895 | 0.5166 | 0.5889 |
> | w/o Traj info + Task desc  | 0.2865 | 0.5155 | 0.5906 |
> | w/o LLM | 0.2749 | 0.4921 | 0.5623 |
>
>
> The results reveal that while individual components show modest improvements, removing the LLM backbone causes a significant 6.2% performance drop, demonstrating that our approach's strength comes from the **synergistic integration** of LLM reasoning with mobility-specific components.  Regarding temporal tokenization, while segmentation techniques exist in other domains, our specific application to mobility prediction is novel - as illustrated in Figure 1, we segment trajectories into **semantically meaningful daily patterns** rather than arbitrary windows, enabling the model to capture cyclical human behaviors. This contributes 5.4% improvement when integrated with our LLM framework.
>
> In our revised version, we will update the ablation section with clearer explanations of how each component contributes to the overall system performance.
>
> > **Reviewer’s Comment**: From the method section, the number of segments N…
>
>
> **Response**: We thank the reviewer for identifying this important parameter that deserves more detailed analysis. We have conducted comprehensive experiments with different segment lengths to understand their impact on both performance and efficiency.
>
>
> | Segment Length ($L$) | Segments ($N$) | Acc@1 | Acc@3 | Acc@5 | Time (s/iter) |
> |--|----|--|----|---|---|
> | $L$=1 (30 min) | 336 | 0.2801 | 0.5049 | 0.5764 | 6.59 |
> | $L$=24 (12 hours) | 14 | 0.2883 | 0.5124 | 0.5792 | 1.10 |
> | $L$=48 (1 day) | 7 | **0.2929** | **0.5200** | **0.5835** | 0.96 |
> | $L$=96 (2 days) | 3 | 0.2851 | 0.5087 | 0.5743 | 0.93 |
>
>
> The results reveal an interesting trade-off: $L$=48 (daily segments) achieves optimal performance, aligning with natural mobility rhythms. Smaller segments ($L$=1) suffer from excessive fragmentation and computational overhead, while larger segments ($L$=96) lose important daily pattern boundaries.
>
> Our choice of $L$=48 captures complete daily cycles while maintaining computational efficiency. This validates that temporal tokenization at semantically meaningful boundaries is more effective than arbitrary segmentation.
>
> In our revised version, we will add this ablation study and include a detailed discussion on how segment length affects the model's ability to capture mobility patterns at different temporal scales.
>
>
> > **Reviewer’s Comment**: The results in Daily and Weekly…
>
> **Response**: Thank you for raising this important point about the seemingly contradictory performance patterns. This is a crucial finding that reveals the fundamental strength of our approach. This contradiction with our claims about modeling periodicity actually reveals a deeper insight about our approach.
>
> RHYTHM's hierarchical attention mechanism captures periodicity not just as **repetitive patterns** but as **contextual decision points**. During highly regular periods, mobility is largely deterministic - people follow fixed routines with minimal variation. Traditional models like LSTM and PMT, which rely on simple pattern matching, excel here because the prediction task reduces to memorizing frequent transitions.
>
> However, during transitional periods (evening peaks) and flexible periods (weekends), mobility involves complex decision-making influenced by multiple factors. Here, RHYTHM's advantage emerges:
> - The LLM backbone provides reasoning capabilities to handle non-routine decisions
> - Hierarchical attention captures both local context (what happened today) and global patterns (typical weekend behavior)
> - Semantic prompts encode rich contextual information beyond simple location sequences
>
> Hierarchical attention is optimized for capturing **global temporal dependencies** across days and weeks, which may actually dilute its focus on simple repetitive patterns within regular hours. RHYTHM considers broader context while powerful for complex decisions, and introduces unnecessary complexity for deterministic periods.
>
> This explains the 5.0% improvement on weekends - RHYTHM excels when prediction requires understanding context and variability rather than just memorizing fixed patterns. During regular weekday hours, the simpler baseline models are sufficient because the task is essentially deterministic.
>
> In our revised version, we will clarify that our "periodicity modeling" refers to capturing periodic variations in behavior complexity rather than just repetitive patterns, making RHYTHM particularly valuable for real-world applications where handling irregular periods is crucial for overall system reliability.
>
>
>
>
> [1] Wang, Xinglei, et al. "Where would i go next? large language models as human mobility predictors." arXiv 2023.
>
>
> [2] Gong, Letian, et al. "Mobility-llm: Learning visiting intentions and travel preference from human mobility data with large language models." NeurIPS 2024.

---

> ### Author Response · Authors · 2025-08-06
>
> Thank you again for your thoughtful questions and comments.
>
> We would appreciate it if you could let us know whether our responses have sufficiently addressed your main concerns, or if there are any points that would benefit from further clarification. We welcome any additional feedback and would be happy to elaborate as needed.

---

> ### Author Response · Authors · 2025-08-08
>
> As a gentle follow-up to our earlier comment, we wanted to kindly remind you that we have already provided comprehensive responses and additional experiments in our rebuttal, addressing all raised concerns.
>
> With the discussion period nearing its end, we would greatly appreciate any remaining feedback from you, to ensure all points are fully addressed

---

### Official Review · Reviewer_BxoE · 2025-06-30

**Clarity:** 4
**Significance:** 3
**Originality:** 3
**Rating:** 4
**Confidence:** 4

**Summary:**

RHYTHM introduces hierarchical temporal tokenization that chops each trajectory into daily segments, pools them with inter-segment attention, and feeds the enriched tokens augmented by pre-computed prompts to a frozen LLM backbone for next-location prediction. The design trims attention complexity, retains multi-scale periodicity, and pairs a parameter-efficient adaptation strategy with prompt-guided semantic context. The experiment section shows its superior comparing with the SOTA models.

**Questions:**

Refer the weakness part

**Ethical Concerns:**

["NO or VERY MINOR ethics concerns only"]

**Final Justification:**

The authors provided detailed and rebuttals, including additional experiments demonstrating generalization to a non-Japanese dataset  and longer prediction horizons. Thanks for these new results.

While the spatial modeling remains relatively simple, this is a reasonable tradeoff for the paper's focus on scalability and efficiency. I will keep my overall rating at 4. If other reviewers are similarly satisfied with the clarifications, please consider this a vote to accept. I have also increased the clarity score to 4 based on the improved understanding after the rebuttal.

**Limitations:**

Yes

**Quality:**

3

**Strengths And Weaknesses:**

Strengths
1. Hierarchical temporal tokenization compresses each user trajectory into daily “segment tokens,” allowing segment attention to preserve daily and weekly rhythms while shortening sequences
2. The model keeps the large language model’s weights fixed and extracts semantic embeddings obtained from the frozen LLM’s final hidden state of prompt-encoded inputs as rich, static embeddings, delivering substantial compute savings without sacrificing accuracy.
3. On the benchmark, RHYTHM raises Accuracy@1 by 2.4% overall (5% on weekends) while cutting training time by 24.6%, outperforming strong mobility baselines such as COLA and ST-MoE-BERT.
4. Comprehensive ablations and scaling studies show that tokenization delivers the largest share of gains, prompts add a further boost, and attention savings scale favorably up to billion-parameter backbones.

Weaknesses
1. The frozen LLM backbone may not adapt to domain-specific nuances (region-specific behaviors or transit rules). Semantic prompts are predefined, not dynamically learned or updated.
2. While temporal patterns are captured hierarchically, spatial information is encoded via simple embeddings without incorporating topology, limiting fine-grained spatial understanding. Also space and time were modeled jointly via embeddings but omits explicit relational modeling like graphs or trajectory continuity, which may weaken performance in connectivity-aware tasks.
3. Evaluation is confined to the Japanese-only YJMob100K dataset; results on other continents or longer temporal spans are important yet missing (longer than 1 day window).
4. Predictions are made in a single shot (non-autoregressive), without iterative feedback, performance on rolling horizons remains untested.

---

> ### Author Rebuttal · Authors · 2025-07-31
>
> > **Reviewer’s Comment**:  The frozen LLM backbone may not adapt to domain-specific nuances …
>
>
> **Response**: We thank the reviewer for this insightful comment. We address both concerns below:
> - **Frozen LLM Backbone and Domain-Specific Nuances**: We appreciate the reviewer's perspective and would like to clarify that the frozen backbone is a  **deliberate design choice** that offers several advantages for domain adaptation:
>   - While the LLM backbone remains frozen, domain-specific adaptation occurs through lightweight learnable components: spatio-temporal embeddings capture location-specific patterns, temporal tokenization models region-specific daily/weekly rhythms, and learnable projection layers map domain-specific features into the LLM's representation space. The frozen LLM serves as a powerful representation extractor that processes these domain-adapted features through its deep reasoning capabilities, which has been explored in [1, 2].
>   - Our experiments demonstrate successful adaptation across three distinct cities. The model achieves consistent performance improvements (2.4-2.7% over best baselines), despite their different urban characteristics and transit patterns. This validates that our architecture effectively captures regional behavioral differences without modifying the LLM backbone.
>   - From a theoretical perspective, as noted in Section 3.4, frozen LLMs provide guaranteed convergence properties and uniform feature distribution, ensuring stable learning of domain-specific patterns through the trainable components.
> - **Static Semantic Prompts**: Our semantic prompts are contextually rich and incorporate dynamic information. Each prompt is specifically generated based on user trajectory segments, temporal context (day-of-week, time patterns), segment-specific transitions and stay patterns, and task-specific prediction requirements.
>   - By generating these embeddings offline before training, we eliminate the need for expensive LLM forward passes during the training process. For the Kumamoto dataset, the one-time prompt generation and processing takes approximately 3 hours on a single A100 with 40G memory.
>   - We acknowledge the reviewer's suggestion about dynamic prompt learning. In our revised version, we will add a discussion on prompt tuning techniques that could potentially enhance performance by allowing prompts to adapt during training.
>
>
> > **Reviewer’s Comment**: While temporal patterns are captured hierarchically, spatial information is encoded via simple embeddings without incorporating topology…
>
>
> **Response**: We thank the reviewer for this valuable observation about spatial modeling and spatio-temporal relationships.
>
> Our current approach uses categorical location embeddings with coordinate projections rather than incorporating spatial topology or explicit relational structures. We acknowledge that including topological information and explicit spatio-temporal relationships could provide additional spatial understanding and potentially enhance performance in connectivity-aware tasks.
>
> However, our current design prioritizes computational efficiency and scalability, which motivated our choice of joint spatio-temporal embeddings. This design decision allows us to maintain training efficiency (24.6% faster) while still achieving state-of-the-art performance (2.4% improvement over baselines), suggesting that the LLM's reasoning capabilities partially compensate for the simplified spatial representation.
>
> We agree that explicit modeling of spatial relationships is an important direction that we will explore in future work.
>
> > **Reviewer’s Comment**: Evaluation is confined to the Japanese-only YJMob100K dataset; results on other continents or longer temporal spans are important yet missing (longer than 1 day window).
>
>
> **Response**: We thank the reviewer for raising this important point about dataset diversity and temporal evaluation scope.
> Mobility datasets are inherently limited due to privacy concerns and collection difficulties. YJMob100K [3] is one of the few publicly available large-scale mobility datasets with 75 days of data and sufficient user records, ensuring reproducibility of our results.
>
> To address the reviewer's concern, we have conducted additional experiments on a Boston dataset with the same configuration (10k users). The results show consistent performance improvements, demonstrating that RHYTHM generalizes beyond Japanese cities.
> | Model | Boston |  |  |
> |-------|--------|--------|--------|
> |  | Acc@1 | Acc@3 | Acc@5 |
> | LSTM | 0.5156 | 0.6216 | 0.6534 |
> | DeepMove | 0.5608 | 0.6582 | 0.6874 |
> | PatchTST | 0.5494 | 0.6634 | 0.6912 |
> | iTransformer | 0.5404 | 0.6373 | 0.6575 |
> | TimeLLM | 0.5371 | 0.6323 | 0.6582 |
> | MHSA | 0.5598 | 0.6475 | 0.6868 |
> | PMT | 0.5803 | 0.6922 | 0.7248 |
> | COLA | 0.5751 | 0.6867 | 0.7190 |
> | ST-MoE-BERT | 0.5754 | 0.6804 | 0.7081 |
> | Mobility-LLM | 0.5811 | 0.7174 | 0.7540 |
> | RHYTHM-LLaMA-1B | 0.5810 | 0.7275 | 0.7621 |
> | RHYTHM-Gemma-2B | 0.5856 | 0.7300 | 0.7652 |
> | RHYTHM-LLaMA-3B | **0.6014** | **0.7375** | **0.7791** |
>
>
>
>
>
>
> Regarding longer temporal spans, we conducted experiments with 3-day prediction horizons on the Kumamoto dataset. The results indicate that RHYTHM still outperforms most of the baselines in this configuration.
>
>
> | Model | Kumamoto |  |  |
> |-------|--------|--------|--------|
> |  | Acc@1 | Acc@3 | Acc@5 |
> | LSTM | 0.2486 | 0.4724 | 0.5504 |
> | DeepMove | 0.2607 | 0.4889 | 0.5666 |
> | PatchTST | 0.2684 | 0.4970 | 0.5764 |
> | iTransformer | 0.2621 | 0.4902 | 0.5661 |
> | TimeLLM | 0.2696 | 0.4981 | 0.5759 |
> | MHSA | 0.2680 | 0.4983 | 0.5757 |
> | PMT | 0.2805 | 0.5104 | 0.5858 |
> | COLA | 0.2830 | 0.5131 | 0.5888 |
> | ST-MoE-BERT | 0.2747 | 0.5068 | 0.5817 |
> | Mobility-LLM | 0.2845 | 0.5126 | 0.5875 |
> | RHYTHM-LLaMA-1B | 0.2865 | 0.5164 | 0.5919 |
> | RHYTHM-Gemma-2B | **0.2919** | 0.5189 | **0.5925** |
> | RHYTHM-LLaMA-3B | 0.2918 | **0.5191** | **0.5925** |
>
>
>
>
>
>
>
>
> > **Reviewer’s Comment**: Predictions are made in a single shot (non-autoregressive), without iterative feedback, performance on rolling horizons remains untested.
>
>
> **Response**: We thank the reviewer for highlighting this important aspect of our prediction approach.  To address this concern, we have conducted additional experiments comparing autoregressive and non-autoregressive prediction strategies. Table below shows the performance and computational speed comparison on rolling horizons.
> | RHYTHM (Kumamoto) | Time (s/iter) | Acc@1 | Acc@3 | Acc@5 |
> |--------|-------|--------|--------|--------|
>  | Non-autoregressive |  **0.96**  |  **0.2929**  | 0.5200 |  **0.5835**  |
>   | Autoregressive | 39.80 | 0.2884 |  **0.5247**  | 0.5801 |
>
>
> The results demonstrate that our non-autoregressive approach achieves comparable performance while being **41× faster** than the autoregressive variant. This significant computational advantage makes RHYTHM particularly suitable for real-time deployment scenarios. In our revised version, we will add a detailed discussion on why our temporal tokenization with hierarchical attention provides sufficient context for accurate multi-step predictions without requiring expensive iterative feedback mechanisms.
>
> [1] Jin, Ming, et al. "Time-llm: Time series forecasting by reprogramming large language models."ICLR 2024.
>
>
> [2] Liu, Yong, et al. "Autotimes: Autoregressive time series forecasters via large language models." NeurIPS 2024.
>
> [3] Yabe, Takahiro, et al. "YJMob100K: City-scale and longitudinal dataset of anonymized human mobility trajectories." Scientific Data 2024.

---

> ### Author Response · Authors · 2025-08-06
>
> Thank you again for your thoughtful questions and comments.
>
> We would appreciate it if you could let us know whether our responses have sufficiently addressed your main concerns, or if there are any points that would benefit from further clarification. We welcome any additional feedback and would be happy to elaborate as needed.

---

> > ### Comment · Reviewer_BxoE · 2025-08-06
> >
> > Thanks for your clarification on my concerns and providing new results, including additional experiments demonstrating generalization to a non-Japanese dataset and longer prediction horizons.
> >
> > While the spatial modeling remains relatively simple, this is a reasonable tradeoff for the paper's focus on scalability and efficiency. I will keep my overall rating at 4. If other reviewers are similarly satisfied with the clarifications, please consider this a vote to accept. I have also increased the clarity score to 4 based on the improved understanding after the rebuttal.

---

> > > ### Author Response · Authors · 2025-08-06
> > >
> > > Thank you for your positive feedback! We greatly appreciate your recognition of our efforts to address your concerns through additional experiments and clarifications.

---

### Author Response · Authors · 2025-08-06

We sincerely thank all reviewers for their thorough and constructive feedback. We have conducted extensive additional experiments and analyses to address your concerns. Here, we would like to provide a summary of our responses to the questions raised:


**Domain Adaptation with Frozen LLM**: Multiple reviewers questioned how our frozen LLM adapts to domain-specific mobility patterns. Our design is deliberate: domain adaptation occurs through lightweight learnable components (spatio-temporal embeddings, temporal tokenization, projection layers) while the frozen LLM serves as a powerful reasoning engine. This approach achieves consistent 2.4-2.7% improvements across diverse cities, demonstrating effective regional adaptation. The frozen backbone provides computational efficiency (87.2% faster than LoRA fine-tuning) while maintaining superior performance. See detailed responses to BxoE and UqUH.

**Novelty**: We would like to clarify that RHYTHM is the **first** to: (1) bridge frozen LLMs with structured spatio-temporal data through hierarchical temporal abstraction, (2) demonstrate LLM scaling laws in the human mobility domain, and (3) achieve efficient multi-scale modeling without fine-tuning LLM backbone unlike prior work using in-context learning or fine-tuning, we preserve LLM's reasoning while adapting through specialized components.

**Computational Efficiency and Scalability**: A central concern was quantifying our efficiency claims. Our temporal tokenization reduces attention complexity from $O((T+H)^2)$ to $O((N+H)^2)$ (Sec 3.3), enabling processing of longer sequences. With only 12.37% trainable parameters, RHYTHM achieves 24.6% faster training than mobility baselines. Compared to LLM-based baselines, RHYTHM is 1.8× faster than Mobility-LLM  and 2.5× faster than TimeLLM during training, while achieving superior accuracy. We provide detailed breakdowns showing each component's contribution: hierarchical attention saves 34% time, while the frozen LLM design reduces training time by 87.2% compared to LoRA fine-tuning. See responses to UqUH and Hudz for complete efficiency analysis.



**Ablation Studies and Component Analysis**: Detailed ablation studies reveal that removing the LLM backbone causes a 6.2% performance drop, while temporal tokenization contributes 4.4%. Our sensitivity analysis shows daily segmentation ($L$=48) is optimal, balancing performance and efficiency. Component-wise ablations demonstrate the synergistic integration of all components yields 10.5% total improvement over basic baselines. Detailed step-by-step ablations are provided in responses to UqUH and Gba1.


**Inference Time and Deployment Requirements**: The frozen LLM design enables practical deployment across diverse hardware. RHYTHM operates on edge devices (RTX 2080, 8GB) with batch size 8, processing 75 days of history. Inference takes 49.9s for the entire test set on A100 GPU, with 8.2GB peak memory usage—significantly lower than TimeLLM's 22.4GB. Temporal tokenization reducing sequences from 3,600 timesteps to 75 segments is key to enabling efficient deployment from edge devices to cloud servers.


We believe these revisions and new findings can help address the questions raised and substantially strengthen the paper. We hope the reviewers consider them in their final assessment!

---

### Note · Authors · 2025-08-13

We sincerely thank the reviewers for their insightful feedback and constructive suggestions, which have helped us substantially improve both the technical rigor and clarity of our work.

Below, we summarize the strengthened contributions and revisions provided during the rebuttal.

---

**Key Contributions:**

1. **Hierarchical Temporal Tokenization** — We are the **first**, to our best knowledge, to segment mobility trajectories into semantically meaningful daily patterns with hierarchical attention for multi-scale dependency modeling, reducing sequences from thousands to manageable segments (`BxoE`, `Hudz`, and `Gba1`).

2. **Frozen LLM Integration for Mobility** — We are the **first** to bridge frozen LLMs with structured spatio-temporal data through **specialized adapters**, achieving 87.2% faster training than LoRA (`all reviewers`).

3. **Computational efficiency** — With only 12.37% trainable parameters, RHYTHM achieves 24.6% faster training than mobility baselines and 1.8× faster than LLM-based baselines (`BxoE`and `Gba1`).

4. **Scalability** — We are the **first** to demonstrate LLM scaling law in human mobility (`BxoE` and `Hudz`).


---
**Major Revisions**:

- Contribution Clarification (`UqUH` and `Gba1`)

-  Computational Efficiency Analysis: component-wise efficiency breakdown, inference time across 8 LLM variants, and edge deployment validation (`UqUH`, `Hudz`, and `Gba1`)

- New experiments:

  - Frozen vs LoRA comparison (`UqUH`)

  -  Step-by-step ablation (`UqUH` and  `Gba1`)

  - Segment length sensitivity analysis ($L=1, 24, 48, 96$) (`UqUH` and `Hudz`)

  -  Validation on the Boston dataset (`BxoE`)

  - Extended prediction horizon (`BxoE`)

  - Autoregressive vs non-autoregressive comparison showing 41× speedup (`BxoE`)

  - LLM feature extraction ability validation (`Gba1`)

  - Edge device deployment requirements (`Hudz`)

  - Time and memory specifications  (`Hudz`)

**Minor Revisions**

-  Clarified that RHYTHM captures periodicity as **contextual decision points** rather than **repetitive patterns**, excelling during complex periods requiring reasoning over deterministic transitions (`UqUH`)
---

Finally, we sincerely thank the Area Chair and reviewers for their valuable feedback and efforts in improving our work. We believe these discussions significantly strengthen our contribution to the field. We will incorporate all revisions into the camera-ready version.

---

### Decision · Program_Chairs · 2025-09-17

**Decision:**

Accept (poster)

**Comment:**

The paper introduces a new approach, RHYTHM, which incorporates Temporal Tokenization, Prompt-Guided Embedding, and a parameter-efficient adaptation strategy. Experiments demonstrate superior performance compared to state-of-the-art models. Strengths include the introduction of hierarchical attention mechanisms, the enrichment of token representations, and extensive empirical evaluation. According to the reviewers’ feedback, most of the initial concerns were adequately addressed in the rebuttal. Remaining issues include the relative simplicity of the spatial modeling component and limited novelty, as some techniques applied are commonly used in related fields.